# Horizon Generalization in Reinforcement Learning

**Vivek Myers**[*]
UC Berkeley
vmyers@berkeley.edu

**Catherine Ji**[*]
Princeton University
cj7280@princeton.edu

**Benjamin Eysenbach**
Princeton University
eysenbach@princeton.edu

## Abstract

We study goal-conditioned RL through the lens of generalization, but not in the traditional sense of random augmentations and domain randomization. Rather, we aim to learn goal-directed policies that generalize with respect to the horizon: after training to reach nearby goals (which are easy to learn), these policies should succeed in reaching distant goals (which are quite challenging to learn). In the same way that invariance is closely linked with generalization is other areas of machine learning (e.g., normalization layers make a network invariant to scale, and therefore generalize to inputs of varying scales), we show that this notion of horizon generalization is closely linked with invariance to planning: a policy navigating towards a goal will select the same actions as if it were navigating to a waypoint en route to that goal. Thus, such a policy trained to reach nearby goals should succeed at reaching arbitrarily-distant goals. Our theoretical analysis proves that both horizon generalization and planning invariance are possible, under some assumptions. We present new experimental results and recall findings from prior work in support of our theoretical results. Taken together, our results open the door to studying how techniques for invariance and generalization developed in other areas of machine learning might be adapted to achieve this alluring property.

## 1 Introduction

Reinforcement learning (RL) is appealing for its potential to use data to solve long-horizon reasoning problems. However, it is precisely this horizon that makes solving the RL problem difficult — the number of possible solutions to a control problem often grows exponentially in the horizon (Kakade, 2003). Indeed, the requirement of collecting long horizon data precludes several potential applications of RL (e.g., health care, robotic manipulation). Thus, a desirable property of an RL algorithm is the ability to learn from short-horizon tasks and generalize to long-horizon tasks. We call this property *horizon generalization.*

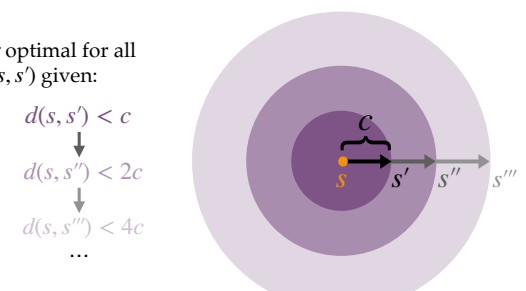

Figure 1: **Horizon generalization.** A policy generalizes over the horizon if performance for start-goal pairs $(s, g)$ separated by a small temporal distance $d(s, g) < c$ yields improved performance over more distant start-goal pairs $(s', g')$ with $d(s', g') > c$.

**Horizon Generalization** (informal statement, see Definition 4): *A goal-conditioned policy generalizes over the horizon if, after training to reach nearby goals, the policy is more successful at reaching distant goals.*

Prior work on generalization in RL almost exclusively focuses on either *(i) perceptual* changes (e.g., changes in lighting conditions), *(ii)* simple randomizations of simulator parameters, or *(iii)* mapping together states and actions with the same reward or value function. While these methods show improved performance on perturbed datasets over the same horizon,

---

Website and code: https://horizon-generalization.github.io
[*]Equal contribution.

they do not generalize over horizon. In this paper, we formalize horizon generalization as a potential property of goal-conditioned RL (GCRL) algorithms, prove that policies with horizon generalization exist, and empirically demonstrate that certain algorithms enable horizon generalization in high-dimensional settings. We do so in the context of goal-conditioned RL, where agents navigate towards a specific goal in a reward-free setting.

A key mathematical tool for understanding horizon generalization is a form of *temporal* invariance obeyed by optimal policies. In the same way that an image classification model that is invariant to rotations will generalize to images of different orientations (Cohen and Welling, 2016; LeCun et al., 2004), we prove that a policy *invariant to planning*, under certain assumptions, will exhibit horizon generalization.

**Planning Invariance** (informal statement, see Definition 3): *A goal-conditioned policy is invariant to planning if it can reach distant goals with similar success when conditioned directly on the goal compared to when conditioned on a series of intermediate waypoints. In other words, breaking up a complex task into a series of simpler tasks confers no advantage.*

The main takeaway from this paper is that there are rich notions of generalization *over the horizon* unique to the goal-conditioned RL (GCRL) setting. We show that existing quasimetric methods (Wang et al., 2023; Myers et al., 2024a) *already* exhibit this form of generalization in high-dimensional settings. By theoretically and empirically linking planning with this form of generalization, our work suggests practical ways (i.e. quasimetric methods) to achieve powerful notions of generalization from short to long horizons.

## 2 RELATED WORK

Our work builds upon prior work in goal-conditioned RL and generalization in RL. Section 5.5 returns to the discussion of prior work in light of our analysis.

**Learning to Reach Goals.** The problem of learning goal-reaching behavior dates to the early days of AI research (Newell, 1959; Laird et al., 1987). This problem has received renewed attention in recent years through the study of deep goal-conditioned reinforcement learning (GCRL) (Chen et al., 2021; Chane-Sane et al., 2021; Colas et al., 2022; Yang et al., 2022; Ma et al., 2022; Schroecker and Isbell, 2020; Janner et al., 2021). Goal-conditioned RL relieves the burden of specifying rewards, as any state in the environment can provide a complete task specification when used as a goal. Some of the excitement in goal-conditioned RL is a reflection of the recent success of self-supervised methods in computer vision (e.g., stable diffusion (Rombach et al., 2022)) and NLP (GPT-4 (OpenAI et al., 2024)): if these methods can achieve intriguing emergent properties (Anil et al., 2022; Brohan et al., 2023), might a self-supervised approach to RL unlock emergent properties for RL?

**Generalization in RL.** Prior work on generalization in RL mostly focuses on variations in *perception* (Cobbe et al., 2019; Stone et al., 2021; Laskin et al., 2020) (or, similarly, e.g., across levels of a game (Nichol et al., 2018; Farebrother et al., 2018; Justesen et al., 2018; Zhang et al., 2018)). Similarly, work on robust RL (which measures a worst-case notion of generalization) usually randomly perturbs the physics parameters (Packer et al., 2018; Eysenbach and Levine, 2022; Moos et al., 2022; Tessler et al., 2019; Igl et al., 2019)). We study a different form of generalization: without changing the dynamics or the observations, can a policy trained on nearby goals succeed in reaching distant goals?

**State Abstractions for Decision-Making.** Many approaches for learning improved state abstractions for decision making have been proposed in recent years, including bisimulation (Ferns et al., 2011; Zhang et al., 2021a; Hansen-Estruch et al., 2022), successor representations (Dayan, 1993; Barreto et al., 2017), and information-theoretic representation learning objectives (Anand et al., 2019; Rakelly et al., 2021; Castro et al., 2021; Jain et al., 2023). While prior work typically views generalization as a problem of handling shift between MDPs with similar horizons, horizon generalization is about generalizing from *short to long horizons*. Prior work that has investigated out-of-distribution, long-horizon tasks has required extra assumptions about the setting, such as access to external planners (Singh et al., 2023; Shah and Levine, 2022; Myers et al., 2024b) or human demonstrations (Mandlekar et al., 2021).

## 3  PLANNING INVARIANCE AND HORIZON GENERALIZATION

Our analysis will focus on the goal-conditioned setting. We first motivate our key formal definitions (planning invariance and horizon generalization), provide preliminaries on quasimetric methods, and prove that these properties can be realized by quasimetric methods.

### 3.1  INTUITION

Many prior works have found that augmenting goal-conditioned policies with planning can significantly boost performance (Savinov et al., 2018; Park et al., 2024): instead of aiming for the final goal, these methods use planning to find a waypoint en route to that goal and aim for that waypoint instead. In effect, the policy chooses a closer, easier waypoint that will naturally bring the agent closer to the final goal.

Invariance to planning (see Fig. 2) is an appealing property for several reasons. First, it implies that the policy realizes the benefits of planning without the complex machinery typically associated with hierarchical and model-based methods. Second, policies optimal over a space of tasks are, by definition, planning-invariant over the same space with respect to an optimal planner: invariance to planning is a necessary but not sufficient condition for policy optimality, and can be used as an inductive bias to achieve policy optimality. Third, we show that planning invariance, combined with other assumptions, implies that the policy will exhibit *horizon generalization*: given that

no planning invariance        planning invariance

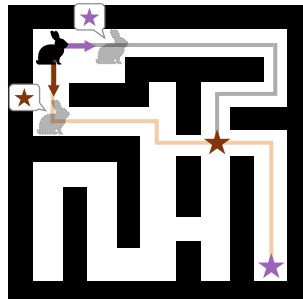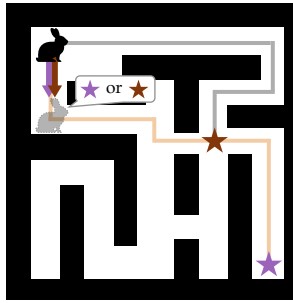

Figure 2: **Visualizing planning invariance.** Planning invariance (Definition 3) means that a policy should take similar actions when directed towards a goal (purple arrow and purple star) as when directed towards an intermediate waypoint (brown arrow and star). We visualize a policy with *(Right)* and without *(Left)* this property via the misalignment and alignment of actions towards the waypoint and the goal, where the optimal path is tan and the suboptimal is gray.

a policy successfully navigates short trajectories covering some state space $\mathcal{S}$, it will succeed at performing long-horizon tasks over the same state space $\mathcal{S}$ (Fig. 1).

How do we actually construct methods that are planning invariant and lead to horizon generalization? To answer this question, we build upon prior work on quasimetric neural network architectures (Liu et al., 2023; Wang and Isola, 2022b;a) and show that policies defined greedily with respect to a quasimetric, where latents obey the triangle inequality, are invariant to planning with respect to the same quasimetric.

## 4  PRELIMINARIES

We consider a controlled Markov process $\mathcal{M}$ with state space $\mathcal{S}$, action space $\mathcal{A}$, and dynamics $p(s' \mid s, a)$. The agent interacts with the environment by selecting actions according to a policy $\pi(a \mid s)$, which is a mapping from $\mathcal{S}$ to distributions over $\mathcal{A}$. We further assume the state and action spaces are compact. We define the *discounted state occupancy measure with actions* as

$$p_\gamma^\pi(\mathfrak{s}_K = g \mid \mathfrak{s}_0 = s, a) \triangleq \sum_{t=0}^{\infty} \gamma^t p^\pi(\mathfrak{s}_t = g \mid \mathfrak{s}_0 = s, a), \tag{1}$$

where $p^\pi(\mathfrak{s}_t = g \mid \mathfrak{s}_0 = s, a)$ is the probability density that policy $\pi$ visits state $g$ after $t$ time steps when initialized at state $s$ with action a.

**Quasimetrics on states.** We equip $\mathcal{M}$ with an additional notion of *distance* between states. We later define planning operator PLAN and policy $\pi$ greedily with respect to this distance. At the most basic level, a distance $d : \mathcal{S} \times \mathcal{S} \to \mathbb{R}$ must be positive for all inputs $(s, s' \neq s)$ and zero for all inputs $(s, s)$ (nonnegativity). We will denote the set of all distances as $\mathcal{D}$:

$$\mathcal{D} \triangleq \{d : \mathcal{S} \times \mathcal{S} \to \mathbb{R} : d(s, s) = 0, d(s, s') > 0 \text{ for each } s, s' \in \mathcal{S} \text{ where } s \neq s'\}. \tag{2}$$

A desirable property for distances to satisfy is the triangle inequality. A distance satisfying this property is known as a *quasimetric*, and we define the set of all quasimetric functions as

$$\mathcal{Q} \triangleq \{d \in \mathcal{D} : d(s,g) \leq d(s,w) + d(w,g) \text{ for all } s,g,w \in \mathcal{S}\}. \tag{3}$$

If we were to restrict the distances to be symmetric ($d(x,y) = d(y,x)$), our quasimetric would become a standard metric obeying nonnegativity, the triangle inequality, and symmetry. However, we wish to use a quasimetric that allows for asymmetry over the interchange of the start and end states: the navigation task $s \to g$ may be completely different from $g \to s$, and the corresponding distance function should reflect this degree of freedom.

An important property of quasimetrics is that they are invariant to the *path relaxation operator* from Dijkstra's algorithm.

**Definition 1** (Path relaxation operator). *Let* $\text{PATH}_d(s,g)$ *be the path relaxation operator over quasimetric* $d(s,g)$. *For any triplet of states* $(s,w,g) \in \mathcal{S} \times \mathcal{S} \times \mathcal{S}$,

$$d(s,g) \leftarrow \text{PATH}_d(s,g) \triangleq \min_w d(s,w) + d(w,g). \tag{4}$$

Thus, invariance to the path relaxation operator is a form of self-consistency. Any triplet of distance predictions should satisfy the following property:

$$d(s,g) \leq d(s,w) + d(w,g)$$

which is the familiar triangle inequality. Quasimetrics naturally satisfy this property and, combined with nonnegativity conditions, are invariant under the path relaxation operator.

**Successor distances (a quasimetric).** A particular quasimetric of note here is the *successor state distance* (Myers et al., 2024a), $d_{\text{SD}}^\gamma$, defined as

$$d_{\text{SD}}^\gamma(s,g) \triangleq \min_\pi \left[ \log \frac{p_\gamma^\pi(\mathfrak{s}_K = g \mid \mathfrak{s}_0 = g)}{p_\gamma^\pi(\mathfrak{s}_K = g \mid \mathfrak{s}_0 = s)} \right], \text{ where } K \sim \text{Geom}(1 - \gamma). \tag{5}$$

The successor distance $d_{\text{SD}}^\gamma$ is a compelling choice of distance because minimizing the distance to the goal $d_{\text{SD}}^\gamma(s,g)$ corresponds to optimal goal reaching with a discount factor $\gamma$.[1] The related *successor distance with actions* $d_{\text{SD}}^\gamma(s,a,g)$ (Myers et al., 2024a) allows us to optimize this distance over actions:

$$d_{\text{SD}}^\gamma(s,a,g) \triangleq \min_\pi \left[ \log \frac{p_\gamma^\pi(\mathfrak{s}_K = g \mid \mathfrak{s}_0 = g)}{p_\gamma^\pi(\mathfrak{s}_K = g \mid \mathfrak{s}_0 = s,a)} \right], \text{ where } K \sim \text{Geom}(1 - \gamma). \tag{6}$$

Given temporal distances between states and state-action pairs, we can define a *quasimetric policy* that greedily selects actions with respect to $d(s,a,g)$:

**Definition 2** (Quasimetric policy). *We define the quasimetric policy as some policy* $\pi_d(a \mid s,g)$ *where*

$$\pi_d(a \mid s,g) \in \underset{a \in \mathcal{A}}{\arg\min}\, d(s,a,g).$$

*Here,* $d(s,a,g)$ *is the successor distance with actions (Eq. 6).*

## 5 Introducing and Analyzing Horizon Generalization

Equipped with these definitions, we can formally define planning invariance and horizon generalization in deterministic and stochastic settings. Then, we show that quasimetric policy $\pi_d(a \mid s,g)$ is planning invariant with respect to a planner defined over the same quasimetric. Finally, we show that this invariance to planning implies horizon generalization. Taken together, our analysis shows that horizon generalization exists and can be achieved by quasimetric methods.

### 5.1 Definitions of Planning Invariance and Horizon Generalization

To construct general definitions of planning invariance and horizon generalization, we will need to define a planning operator which proposes waypoints at a given state to reach a target distribution over goals.

---

[1]Formally, define an MDP with the goal-conditioned reward function $r_g(s) = \delta_{(s,g)}$, a Kronecker delta function which evaluates to 1 if $s = g$ and 0 otherwise. The $d_{\text{SD}}^\gamma$-minimizing policy is optimal for this MDP.

We denote by

$$\mathbf{plan} \triangleq \{\text{PLAN} : \mathcal{S} \times \Delta(\mathcal{S}) \mapsto \Delta(\mathcal{S})\} \tag{7}$$

the class of *planning functions* that predict a distribution of waypoints from states and goals. In the special case of deterministic actions, waypoints, and goals, we write

$$\mathbf{plan}^{\text{FIX}} \triangleq \{\text{PLAN}^{\text{FIX}} : \mathcal{S} \times \mathcal{S} \mapsto \mathcal{S}\} \subset \mathbf{plan}. \tag{8}$$

Our analysis in the rest of this section will focus on the simpler "fixed" setting of $\text{PLAN}^{\text{FIX}} \in \mathbf{plan}^{\text{FIX}}$. We will use $w$ or $w_{\text{PLAN}}$ to denote the waypoint produced by $\text{PLAN}^{\text{FIX}}(s, g)$. The proofs and quasimetric objects in the stochastic setting are slightly more complicated, but carry the same structure and takeaways as this simpler case; the general stochastic proofs and definitions are presented in Appendix B. For notational brevity, we drop the label $^{\text{FIX}}$ in the rest of the analysis section.

There are many possible planning algorithms one could use (e.g., Dijkstra's algorithm (Dijkstra, 1959), A* (Hart et al., 1968), RRT (LaValle and Kuffner, 2001)). Importantly, the constraints of a quasimetric (see Section 4) and the related idea of *path relaxations* from Dijkstra's algorithm provide clues for specifying our planning operator later in our analysis. We use this planning operator in one of our key definitions (visualized in Fig. 2):

**Definition 3** (Planning invariance). *Consider a deterministic MDP with states $\mathcal{S}$, actions $\mathcal{A}$, and goal-conditioned Kronecker delta reward function $r_g(s) = \delta_{(s,g)}$. For any goal-conditioned policy $\pi(a \mid s, g)$ where $g \in \mathcal{S}$, we say that $\pi(a \mid s, g)$ is invariant under planning operator $\text{PLAN} \in \mathbf{plan}$ if and only if*

$$\pi(a \mid s, g) = \pi(a \mid s, w), \text{ where } w = \text{PLAN}(s, g). \tag{9}$$

Note that planning invariance says nothing about whether the planner is good or bad. We will primarily be interested in invariance under the optimal planner with respect to some quasimetric $d(s, g)$. We denote the class of quasimetric planning functions as

$$\mathbf{plan}_d \triangleq \{\text{PLAN} \in \mathbf{plan} \mid d(s, \text{PLAN}(s, g)) + d(\text{PLAN}(s, g), g) = d(s, g) \text{ for all } (s, g) \in \mathcal{S} \times \mathcal{S}\}.$$

Our second key definition is horizon generalization (see Fig. 1):

**Definition 4** (Horizon generalization). *Suppose $c > 0$ and $d(s, g)$ is a quasimetric over the start-goal space $\mathcal{S} \times \mathcal{S}$. In the single-goal, controlled ("fixed") case, a policy $\pi(a \mid s, g)$ **generalizes over the horizon** if optimality over nearby start-goal pairs $\mathcal{B}_c = \{(s, g) \in \mathcal{S} \times \mathcal{S} \mid d(s, g) < c\}$ everywhere implies optimality over the entire state space $\mathcal{S}$.*

We highlight the key base case assumption: optimality over shorter trajectories that *cover the entire desired state space* $\mathcal{S}$ leads to horizon generalization. We assume this base case holds *everywhere* — without additional assumptions about the symmetries of the MDP, it is beyond the scope of this work to consider horizon generalization to completely unseen states. Rather, we analyze generalization to unseen, long-horizon $(s, g)$ state *pairs*.

## 5.2 QUASIMETRIC POLICIES ARE (NONTRIVIALLY) PLANNING INVARIANT

With these notions of planning invariance and horizon generalization in hand, we will consider nontrivial quasimetric planning algorithms $\text{PLAN}_d \in \mathbf{plan}_d$ that acquire a quasimetric $d(s, g)$ and output a single waypoint $w \in \mathcal{S}$:

$$\text{PLAN}_d(s, g) = w_{\text{PLAN}} \in \arg\min_{w \in \mathcal{S}} d(s, w) + d(w, g). \tag{10}$$

Note that this planning algorithm takes the form of the path relaxation operator (Definition 1). By the triangle inequality, we have $d(s, w_{\text{PLAN}}) + d(w_{\text{PLAN}}, g) = d(s, g)$. Our first result is that quasimetric policies $\pi_d$ are invariant under planning operator $\text{PLAN}_d$.

**Theorem 1** (Quasimetric policies are invariant under $\text{PLAN}_d$). *Given a deterministic MDP with states $\mathcal{S}$, actions $\mathcal{A}$, and goal-conditioned Kronecker delta reward function $r_g(s) = \delta_{(s,g)}$, define quasimetric policy $\pi_d(a \mid s, g)$ and quasimetric planner class $\mathbf{plan}_d$. Then, for every quasimetric planner $\text{PLAN}_d \in \mathbf{plan}_d$, there always exists a policy $\pi_d(a \mid s, g)$ that is planning invariant:*

$$\pi_d(a \mid s, g) = \pi_d\big(a \mid s, w \text{ for } w = \text{PLAN}_d(s, g)\big). \tag{11}$$

The proof is in Appendix B.1. In practice, we measure planning invariance by comparing the relative *performance* of algorithms with and without planning. For this condition, we do not necessarily need $\pi_d(a \mid s, g) = \pi_d(a \mid s, w_{\text{PLAN}})$; rather, the weaker condition $d(s, \pi_d(a \mid s, g), g) = d(s, \pi_d(a \mid s, w_{\text{PLAN}}), w_{\text{PLAN}})$ is sufficient and necessary for planning invariance when there are no errors from function approximation and noise. We extend this result to stochastic settings in Appendix B.3.

## 5.3 Quasimetric Policies Generalize over the Horizon

Our main result of this section is to prove that horizon generalization exists for quasimetric policies through an inductive argument.

**Theorem 2** (Horizon generalization exists)**.** *Consider an MDP with states $\mathcal{S}$, actions $\mathcal{A}$, and goal-conditioned Kronecker delta reward $r_g(s) = \delta_{(s,g)}$. For any quasimetric $d(s, g)$ over the start-goal space $\mathcal{S} \times \mathcal{S}$ and $c > 0$, the quasimetric policy $\pi_d(a \mid s, g)$ that is optimal over $\mathcal{B}_c = \{(s, g) \in \mathcal{S} \times \mathcal{S} \mid d(s, g) < c\}$ is optimal over the entire start-goal space $\mathcal{S} \times \mathcal{S}$.*

The idea of the proof is to begin with a ball of states $\mathcal{B}_c(s) = \{s' \in \mathcal{S} \mid d(s, s') < c\}$ on which the policy $\pi_d(a \mid s, \cdot)$ is optimal, and show that planning invariance and the triangle inequality in turn imply optimality over a ball of states with double the radius $\mathcal{B}_{2c}(s)$.

Thus, there must exist planning-invariant goal-reaching policies that generalize over the horizon: optimality over pairs of close states everywhere implies optimality over arbitrarily distant pairs of states. The complete proof, extended to stochastic settings, is in Appendix B.4.

We also observe that horizon generalization is not guaranteed for a goal-reaching policy that is *not* planning invariant (Remark 3, see Appendix B.5 for proof).

**Remark 3** (Horizon generalization is nontrivial)**.** *Let finite $c > 0$ and goal-conditioned MDP with states $\mathcal{S}$, actions $\mathcal{A}$, and goal-conditioned Kronecker delta reward function $r_g(s) = \delta_{(s,g)}$ be given where there are no states outside of $\mathcal{S}$. For a policy that is not planning invariant, optimality over $\mathcal{B}_c = \{(s, g) \in \mathcal{S} \times \mathcal{S} \mid d(s, g) < c\}$ is not a sufficient condition for optimality over the entire start-goal space $\mathcal{S} \times \mathcal{S}$.*

Combined, these results show that planning invariance and horizon generalization, as defined in Section 5.1, exist in nontrivial forms via quasimetric policies.

## 5.4 Limitations and Assumptions

Despite our theoretical results proving that horizon generalization exists, we expect that practical algorithms will not *perfectly* achieve horizon generalization.

An assumption in our inductive proof is that horizon generalization exists as a binary category. However, in practice, each application of the inductive argument will incur some error, so the extent of horizon generalization will not be indefinite.

Concretely, define $\text{SUCCESS}(c)$ as the success rate for reaching goals in radius $c$, and assume that we choose $c_0$ small enough that $\text{SUCCESS}(c_0) = 1$. Suppose each time the horizon is doubled $(c_0 \to 2c_0 \to 4c_0 \to \cdots)$, the success rate decreases by a factor of $\eta$. We will refer to $\eta$ as the **horizon generalization parameter** and later measure this parameter in our experiments (Section 6). We can now define the REACH as the sum of $\text{SUCCESS}(c)$ over $c \geq c_0$. With the above constraints on $\text{SUCCESS}(c)$, in the worst case,

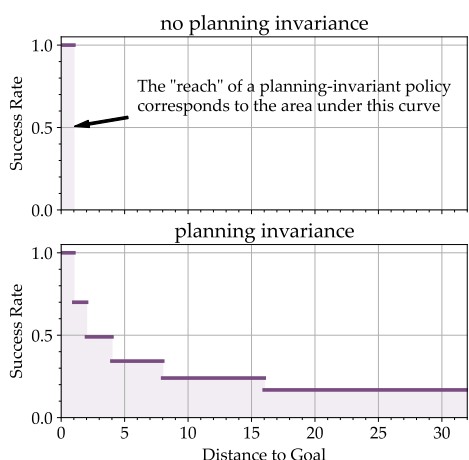

Figure 3: ***Approximate* horizon generalization is still useful.** SUCCESS when there is horizon generalization. When the success attenuation factor $\eta \geq 0.5$, the REACH goes to $\infty$. For a policy with no horizon generalization ($\eta = 0$), its REACH = 1.

$$\text{REACH}_{wc} = 1 + \eta(2 - 1) + \eta^2(4 - 2) + \eta^3(8 - 4) + \cdots = \begin{cases} 1 + \eta \frac{1}{1 - 2\eta} & \text{if } 0 < \eta < 1/2 \\ \infty & \text{if } \eta \geq 1/2 \end{cases} . \quad (12)$$

When there is no horizon generalization, the Reach is 1. We can see this by integrating the success curve in Fig. 3. When the degree of horizon generalization has a low value of (say) $\eta = 0.1$ (i.e., it generalizes for only 1 out of every 10 goals), the Reach is 1.125, not much bigger than that of a policy without horizon generalization. Once the degree of horizon generalization reaches $\eta = 1/2$ (i.e., generalizes for 1 out of every two goals), the Reach is infinite. In short, the potential reach of horizon generalization is infinite, even when each step of the recursive argument incurs a non-negligible degree of error.

A second assumption behind our analysis is that the base case holds *everywhere*: the policy must succeed at reaching *all* nearby goals when initialized at *all* possible starting states. In practice, this may translate to a coverage assumption on the training data. If the base case does not hold (poor performance on easy goals) but planning invariance holds, then we do not expect to see optimality over arbitrarily hard goals. We observe this empirically in our experiments (Fig. 4): a random policy is invariant to planning (it always selects random actions, regardless of the goal) yet its performance on nearby goals is poor, so the policy fails to exhibit horizon generalization.

Finally, note that invariance under any arbitrary planner does not guarantee horizon generalization. Rather, only invariance under a planner *that minimizes an asymmetric distance* (quasimetric) leads to horizon generalization.

Nonetheless, planning invariance is attractive for several reasons: (1) planning-invariant policies potentially automatically get the benefits of planning, (2) optimal policies are invariant under optimal planners, and, as we show in our analysis, (3) invariance to planners that shorten quasimetric distances leads to horizon generalization (Theorem 2).

### 5.5 WHICH PRACTICAL METHODS MIGHT EXHIBIT HORIZON GENERALIZATION?

Temporal difference methods (Ziebart et al., 2008), quasimetric architectures (Wang et al., 2023), RL algorithms (Savinov et al., 2018), and data augmentations (Ghugare et al., 2024) that employ explicit planning can all achieve planning invariance under some assumptions.

Appendix C discusses these methods in more detail. In Appendix C.1, we discuss several new directions for designing RL algorithms that are invariant to planning. Appendix C.2 recalls figures from prior works in search of evidence for horizon generalization.

## 6 EXPERIMENTS

The aim of our experiments is to demonstrate horizon generalization and planning invariance in existing RL settings, and to study the extent to which existing methods already achieve these properties. We also present an experiment highlighting why horizon generalization is a useful notion even when considering temporal difference methods (Section 6.2).

We start with a didactic, tabular navigation task (Fig. 11), connecting short horizon trajectories and evaluating performance on long-horizon tasks. In our first experiment, we measure the empirical average hitting time distance between all pairs of states. We define a policy that acts greedily with respect to these distances, measuring performance of this "metric regression" policy in Fig. 4 *(Top Left)*. The degree of horizon generalization can be quantified by comparing its success rate on nearby $(s, g)$ pairs to more distant pairs. We compare to a "metric regression with quasimetric" method that projects the empirical hitting times *into a quasimetric* by performing path relaxation updates until convergence $(d(s, g) \leftarrow \min_w d(s, w) + d(w, g))$. Fig. 4 *(Top Left)* shows that this policy achieves near perfect horizon generalization. While this result makes intuitive sense (this algorithm is very similar to Dijkstra's algorithm), it nonetheless highlights one way in which a method trained on nearby start-goal pairs can generalize to more distant pairs.

We study planning invariance of these policies by comparing the success rate of each policy (on distant start-goal pairs) when the policy is conditioned on the goal versus on a waypoint. See Appendix E for details. As shown in Fig. 4 *(Top Right)*, the "metric regression with quasimetric" policy exhibits stronger planning invariance, supporting our theoretical claim that (Theorem 1) planning invariance is possible.

We next study whether these properties exist when using function approximation. For this experiment, we adopt the contrastive RL method (Eysenbach et al., 2022) for estimating

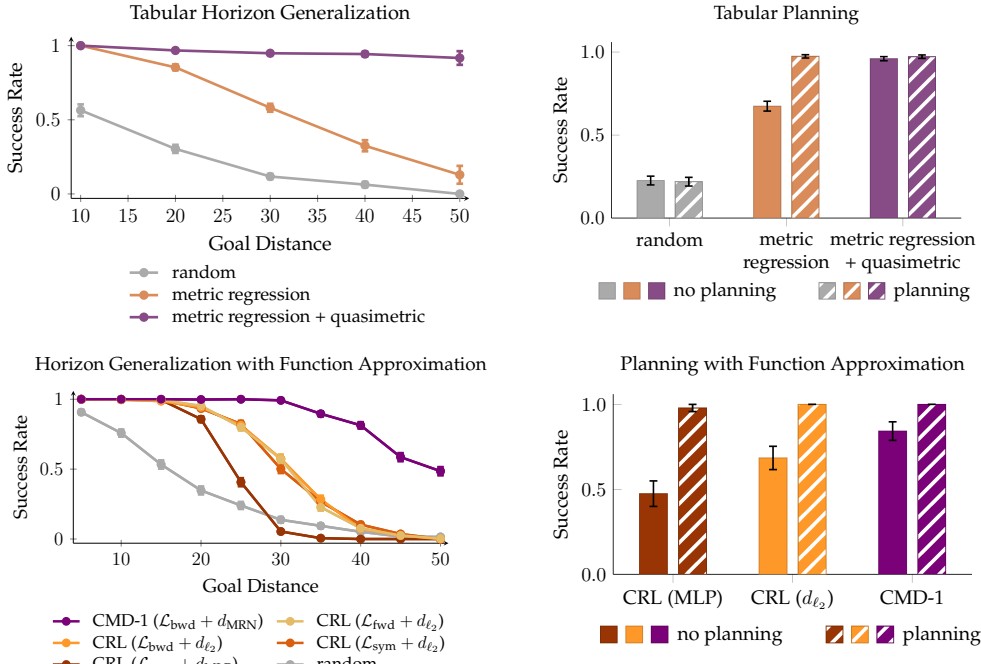

Figure 4: **Quantifying horizon generalization and invariance to planning.** On a simple navigation task, we collect short trajectories and train two goal-conditioned policies, comparing both to a random policy. *(Top Left)* We evaluate on $(s, g)$ pairs of varying distances, observing that metric regression with a quasimetric exhibits strong horizon generalization. *(Top Right)* In line with our analysis, the policy that has strong horizon generalization is also more invariant to planning: combining that policy with planning does not increase performance. *(Bottom Row)* We repeat these experiments using function approximation (instead of a tabular model), observing similar trends.

the distances, comparing different architectures and loss functions. The results in Fig. 4 *(Bottom Left)* show that both the architecture and the loss function can influence horizon generalization, with the strongest generalization being achieved by a CMD-1 (Myers et al., 2024a). Intuitively this makes sense, as this method was designed to exploit the triangle inequality, which is linked to planning invariance. Fig. 4 *(Bottom Right)* shows the degree of planning invariance for these policies. Supporting our analysis, the policy most invariant to planning when trained on short horizon tasks shows the strongest horizon generalization.

To better understand the relationship between planning invariance and horizon generalization, we used the data from Fig. 4 *(Bottom Left)* to estimate the horizon generalization parameter $\eta$ (see Section 5.4), and used the data from the *(Bottom Right)* to compute the ratio of performance with and without planning (Fig. 7). These two quantities are well correlated, supporting Theorem 2's claim that horizon generalization is closely linked to planning invariance. Methods that use an L2-distance parameterized architecture showed stronger horizon generalization and planning invariance than that which uses an MLP, suggesting that some degree of planning

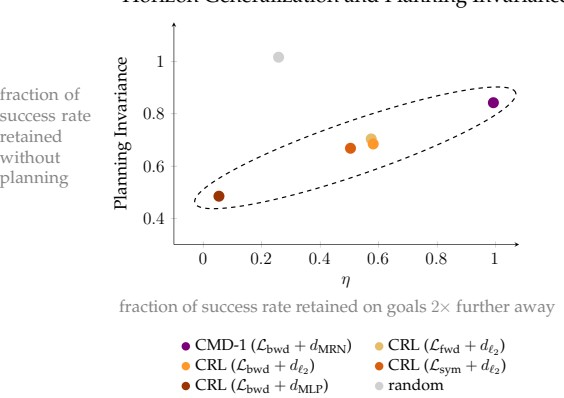

Figure 7: Relating horizon generalization ($x$-axis) to planning invariance ($y$-axis).

invariance is possible even without a quasimetric architecture. Intriguingly, methods using the L2 architecture have a value of $\eta \approx 0.5$, right at the critical point between bounded and unbounded reach (see Section 5.4). The CMD-1 method, which is explicitly designed to

(a) Ant Environment    (b) Success rates stratified by distance to goal

Figure 5: **Measuring horizon generalization in a high-dimensional (27D observation, 8DoF control) task.** *(Left)* We use an enlarged version of the quadruped "ant" environment, training all goal-conditioned RL methods on (start, goal) pairs that are at most 10 meters apart. *(Right)* We evaluate several RL methods, measuring the horizon generalization of each. These results reveal that *(i)* some degree of horizon generalization is possible; *(ii)* the learning algorithm influences the degree of generalization; *(iii)* the value function architecture influences the degree of generalization; and *(iv)* no method achieves perfect generalization, suggesting room for improvement in future work. The ratio of success at 10m vs 5m and 20m vs 10m corresponds to $\eta$ from Section 5.4.

(a) AntMaze    (b) Humanoid

| | $\eta$ value | |
|---|---|---|
| Distance | CMD | CRL |
| 5m | **1.00** | 1.00 |
| 15m | **0.93** | 0.10 |
| 25m | **0.25** | 0.00 |

| | $\eta$ value | |
|---|---|---|
| Distance | CMD | CRL |
| 5m | **1.00** | 1.00 |
| 15m | **0.69** | 0.60 |
| 25m | **0.24** | 0.16 |

Figure 6: **Illustrating Horizon Invariance in Additional Environments.** *(Left)* A large Ant maze environment with a winding S-shaped corridor. *(Right)* A humanoid environment with a complex, high-dimensional observation space. We evaluate the horizon generalization as measured by $\eta$ for a quasimetric architecture (CMD) and a standard architecture (CRL), quantifying the ratio of success rates when evaluating at 5m vs 10m, 15m vs 30m, and 25m vs 50m after training to reach goals within 10m. The largest $\eta$ values in each row are **highlighted**.

incorporate the triangle inequality, exhibits much stronger planning invariance and horizon generalization ($\eta \approx 0.8 \gg 0.5$), well above the critical point.

Note that the random policy is an outlier: it achieves perfect planning invariance (it always takes random actions, regardless of the goal) yet poor horizon generalization. This random policy highlights a key assumption in our analysis: that the policy *always* succeeds at reaching nearby goals (in Fig. 4, note that the success rate on the easiest goals is strictly less than 1).

## 6.1 STUDYING HORIZON GENERALIZATION IN A HIGH-DIMENSIONAL SETTING

Our next set of experiments study horizon generalization and planning invariance in the context of a high-dimensional quadrupedal locomotion task (see Fig. 5). We start by running a series of experiments to compare the horizon generalization of different learning algorithms (CRL (Eysenbach et al., 2022) and SAC (Haarnoja et al., 2018)) and distance metric architectures (details in Appendix E). The results in Fig. 5 highlight that both the learning algorithm and the architecture can play an important role in horizon generalization, while also underscoring that achieving high horizon generalization in high-dimensional settings remains an open problem. See Table 1 for a summary of the methods used in these experiments.

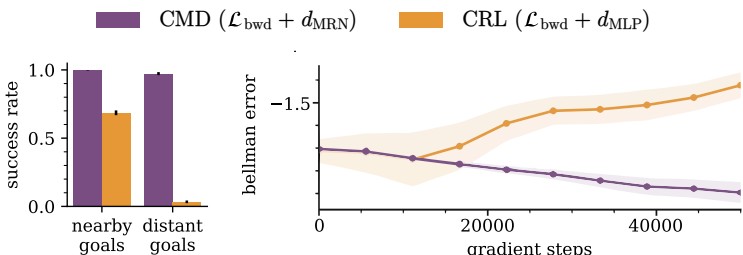

Figure 8: **Impact of horizon generalization on Bellman errors.** *(Left)* Two goal-reaching methods exhibit different horizon generalization. *(Right)* Despite neither method being trained with the Bellman loss, we observe that the method with stronger horizon generalization has a lower Bellman loss. Thus, understanding horizon generalization may be important even when using TD methods (which guarantee horizon generalization at convergence).

These trends hold in more complex environments as well: Fig. 6 shows greater horizon generalization (as measured by the $\eta$-value defined in Section 5.4) for a CMD-1 architecture compared to a CRL architecture in both an AntMaze and a Humanoid environment.

### 6.2 IMPACT OF HORIZON GENERALIZATION ON BELLMAN ERRORS

Why should someone using a temporal difference method care about horizon generalization, if TD methods are supposed to provide this property for free? One hypothesis is that methods for achieving horizon generalization will also help decrease the Bellman error, especially for unseen start-goal pairs. We test this hypothesis by measuring the Bellman error throughout training of the contrastive RL method (same method as Fig. 4), with two different architectures. The results in Fig. 8 show that the architecture that exhibits stronger horizon generalization $(d_{\ell_2})$ also has a lower Bellman error throughout training. Thus, while TD methods may achieve horizon generalization at convergence (at least in the tabular setting with infinite data), a stronger understanding of horizon generalization may nonetheless prove useful for designing architectures that enable faster convergence of TD methods.

## 7 CONCLUSION

The aim of this paper is to give a name to a type of generalization that has been observed before, but (to the best of our knowledge) has never been studied in its own right: the capacity to generalize from nearby start-goal pairs to distant goals. Seen from one perspective, this property is trivial — it is an application of the optimal substructure property, and the original Q-learning method (Watkins and Dayan, 1992) already achieves this property. Seen from another perspective, this property may seem magical: how can one *guarantee* that a policy trained over easy tasks can *extrapolate* from easy tasks to hard tasks?

We hope to provide a theoretical framework for understanding this property as a form of self-consistency over model architecture, and show how we can obtain and measure this property in practice. The experiments in Section 6 then connect these insights to concrete advice for structuring the representation for goal-reaching:

1. Policies that model state distance with metric architectures have *planning invariance*.
2. Planning invariance is correlated with the degree of *horizon generalization* of a policy.
3. Quasimetric architectures can enable planning invariance and horizon generalization.

Appendix D discusses further implications of these notions of invariance on model consistency.

**Limitations and Future Work.** Future work should examine planning invariance and horizon generalization in more complex decision-making environments, such as robotic manipulation and language-based agents. Which versions of the distance parameterizations in Table 1 are most effective at scale should be investigated with larger-scale empirical experiments. We assume a goal-conditioned setting, but there are meany alternative forms of task specification (rewards, language, preferences, etc.) that could similarly benefit from generalization over long horizons. Future work should explore how planning-invariant geometry could be extended or mapped onto these task spaces.

ACKNOWLEDGMENTS

We thank Cindy Zhang, Ryan Adams, and Seohong Park for helpful discussions. We thank Princeton Research Computing for assistance with the numerical experiments. We also thank anonymous reviewers who have helped improve the paper.

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

## A    DEFINITION OF PATH RELAXATION

In this section, we formally define the general path relaxation operator in Definition 5. This definition extends Definition 1 to allow for actions and environmental stochasticity.

**Definition 5** (Path relaxation operator with actions). *Let* $\text{PATH}_d(s, a, G)$ *be the path relaxation operator over quasimetric* $d(s, a, G)$. *For any triplet of state and state distributions* $(s, W, G) \in \mathcal{S} \times \Delta(\mathcal{S}) \times \Delta(\mathcal{S})$,

$$\text{PATH}_d(s, a, G) \triangleq \min_W d(s, a, W) + d(W, G). \tag{13}$$

*In the controlled, fixed goal setting, define*

$$\text{PATH}_d^{FIX}(s, a, g) \triangleq \min_w d(s, a, w) + d(w, g). \tag{14}$$

The notation $\Delta(X)$ used here and throughout the appendix denotes the space of probability distributions over set $X$.

## B    FORMALIZING PLANNING INVARIANCE AND HORIZON GENERALIZATION

In this section, we prove results discussed in Section 5.2 and versions of results in Section 5 for the general stochastic, distributional setting.

### B.1    PLANNING INVARIANCE EXISTS

**Theorem 1** (Quasimetric policies are invariant under $\text{PLAN}_d$). *Given a deterministic MDP with states* $\mathcal{S}$, *actions* $\mathcal{A}$, *and goal-conditioned Kronecker delta reward function* $r_g(s) = \delta_{(s,g)}$, *define quasimetric policy* $\pi_d(a \mid s, g)$ *and quasimetric planner class* $\textbf{plan}_d$. *Then, for every quasimetric planner* $\text{PLAN}_d \in \textbf{plan}_d$, *there always exists a policy* $\pi_d(a \mid s, g)$ *that is planning invariant:*

$$\pi_d(a \mid s, g) = \pi_d\big(a \mid s, w \text{ for } w = \text{PLAN}_d(s, g)\big). \tag{11}$$

*Proof.* Let $s, g \in \mathcal{S}$ and the action-free distance function be $d(s, g) = \min_a d(s, a, g)$; this statement is true for the contrastive successor distances (Eq. 5). Define the (deterministic) planned waypoint as

$$w_{\text{PLAN}} \leftarrow \text{PLAN}_d(s, g) \in \underset{w \in \mathcal{S}}{\arg\min} \, d(s, w) + d(w, g). \tag{15}$$

We can then construct the following policy:

$$\pi_d(a \mid s, g) \in \underset{a \in \mathcal{A}}{\arg\min} \, d(s, a, g) \tag{16}$$

and later restrict the selection of equivalently optimal actions to obtain planning invariance, where $w_{\text{PLAN}} \in \arg\min_{w \in \mathcal{S}} d(s, w) + d(w, g)$. Applying this policy to $(s, w_{\text{PLAN}})$, we have that

$$\pi_d(a \mid s, w_{\text{PLAN}}) \in \underset{a \in \mathcal{A}}{\arg\min} \, d(s, a, w_{\text{PLAN}})$$

$$= \underset{a \in \mathcal{A}}{\arg\min} \, d(s, a, w_{\text{PLAN}}) + d(w_{\text{PLAN}}, g)$$

$$= d(s, w_{\text{PLAN}}) + d(w_{\text{PLAN}}, g)$$

$$\subseteq \underset{a \in \mathcal{A}}{\arg\min}\, d(s, a, g). \qquad (17)$$

Thus, for a given deterministic planning algorithm defined as in Eq. (15), there exists some deterministic policy $\pi_d(a \mid s, g) = \pi_d(a \mid s, w_{\text{PLAN}}) \in \arg\min_{a \in \mathcal{A}} d(s, a, w_{\text{PLAN}}) \subseteq \arg\min_{a \in \mathcal{A}} d(s, a, g)$ that is planning invariant. $\qquad \square$

### B.2 Quasimetric Over Distributions

**Definition 6** (Quasimetric over distributions). *Given quasimetric $d_{\text{QM}}$ defined over start-goal space $\mathcal{S} \times \mathcal{S}$, we define the quasimetric over distributions as*

$$d_{\text{QMD}}(P, Q) = \inf_{\gamma \in \Pi(P, Q)} \int_{\mathcal{S} \times \mathcal{S}} d_{\text{QM}}(p, q)\gamma(p, q)\, \mathrm{d}p\, \mathrm{d}q, \qquad (18)$$

*which is the asymmetric Wasserstein Distance with quasimetric cost function $d_{QM}(p, q)$.*

We can interpret this object as the minimum cost to convert distribution $P$ to $Q$, where the cost function is some quasimetric between individual states.

We show Definition 6 is a valid quasimetric. Because $d_{\text{QMD}}$ is an asymmetric Wasserstein distance and cost function $d_{\text{QM}}(p, q)$ is a quasimetric, this proof is an extension of a well-known result (Clement and Desch, 2008) that drops the metric symmetry condition. We include the proof here for completeness.

*Proof.* We check the conditions of a quasimetric for $d_{\text{QMD}}(P, Q)$ with quasimetric cost function $d_{\text{QM}}(p, q)$.

**Non-negativity:** By definition of $\gamma(p, q)$ and $d_{\text{QM}}(p, q)$, we have $d_{\text{QMD}}(P, Q) \geq 0$ for all $P, Q$.

We show that $d_{\text{QMD}}(P, Q) = 0$ if and only if $P = Q$, beginning with the forward direction:

$$
\begin{aligned}
d_{\text{QMD}}(P, P) &= \inf_{\gamma \in \Pi(P, P)} \int_{\mathcal{S} \times \mathcal{S}} d_{\text{QM}}(p, q)\gamma(p, q)\ \mathrm{d}p\ \mathrm{d}q \\
&\leq \int_{\mathcal{S} \times \mathcal{S}} d_{\text{QM}}(p, q)\gamma_D(p, q)\ \mathrm{d}p\, \mathrm{d}q && \text{(set } \gamma \text{ as diagonal matrix } \gamma_D) \\
&= \int_{\mathcal{S}} d_{\text{QM}}(p, p)\mu(p)\ \mathrm{d}p && \text{(where } \mu(p) = \gamma_D(p, p)) \\
&= 0 && (d_{QM}(p, p) = 0)
\end{aligned}
$$

For the other direction, we have that $d_{\text{QMD}}(P, Q) = 0$ implies $\gamma(p, q) = 0$ for all $p \neq q$. However, because $\gamma(p, q)$ is a probability distribution, this must mean $P = Q$.

**Asymmetry:** We have that $d_{\text{QMD}}(P, Q)$ is not necessarily symmetric because the quasimetric $d_{\text{QM}}(p, q)$ is not necessarily symmetric.

**Triangle inequality:** Let $P, Q, R$ be three probability measures. Let $\gamma_{1,2}^*$ and $\gamma_{2,3}^*$ be minimizers of $d_{\text{QMD}}(P, Q)$ and $d_{\text{QMD}}(Q, R)$ respectively. We can construct some $\gamma_{1,2,3}(p, q, r)$ such that

$$\int_{\mathcal{S}} \gamma_{1,2,3}(p, q, r)\, \mathrm{d}r = \gamma_{1,2}^*$$

$$\int_{\mathcal{S}} \gamma_{1,2,3}(p, q, r)\, \mathrm{d}p = \gamma_{2,3}^*$$

$$\int_{\mathcal{S}} \gamma_{1,2,3}(p, q, r)\, \mathrm{d}q = \gamma_{1,3}$$

where $\gamma_{1,3}$ is **not necessarily** the optimal joint distribution to minimize $d_{\text{QMD}}(P, R)$. Then, we have:

$$
\begin{aligned}
d_{\text{QMD}}(P, R) &\leq \int_{\mathcal{S} \times \mathcal{S}} d_{\text{QM}}(p, r)\gamma_{1,3}(p, r)\, \mathrm{d}p\ \mathrm{d}r \\
&= \int_{\mathcal{S} \times \mathcal{S} \times \mathcal{S}} d_{\text{QM}}(p, r)\gamma_{1,2,3}(p, q, r)\, \mathrm{d}p\ \mathrm{d}q\ \mathrm{d}r
\end{aligned}
$$

$$\leq \int_{\mathcal{S} \times \mathcal{S} \times \mathcal{S}} (d_{\mathrm{QM}}(p,q) + d_{\mathrm{QM}}(q,r)) \, \gamma_{1,2,3}(p,q,r) \, \mathrm{d}p \, \mathrm{d}q \, \mathrm{d}r \qquad (d_{\mathrm{QM}} \text{ satisfies } \triangle\text{-ineq})$$

$$= \int_{\mathcal{S} \times \mathcal{S} \times \mathcal{S}} d_{\mathrm{QM}}(p,q) \gamma_{1,2,3}(p,q,r) \, \mathrm{d}p \, \mathrm{d}q \, \mathrm{d}r + \int_{\mathcal{S} \times \mathcal{S} \times \mathcal{S}} d_{\mathrm{QM}}(q,r) \gamma_{1,2,3}(p,q,r) \, \mathrm{d}p \, \mathrm{d}q \, \mathrm{d}r$$

$$= d_{\mathrm{QMD}}(P,Q) + d_{\mathrm{QMD}}(Q,R)$$

as desired. Therefore, $d_{\mathrm{QMD}}$ is a quasimetric and we are done. $\qquad\square$

### B.3 Quasimetrics, Path Relaxation, Policies, and Planning Invariance in Stochastic Settings

We extend the definitions of quasimetrics, path relaxation, policies, and planning invariance to the setting of general stochastic MDPs.

**Definition 7** (Quasimetric over actions in general stochastic setting). *Denote by $S'_{s,a} = p(s' \mid s,a)$ the distribution over next-step states after taking action $a$ from starting state $s$. Given the quasimetric $d_{\mathrm{QMD}}$ defined as in Definition 6, we define the stochastic-setting quasimetric over actions as*

$$d_{\mathrm{QMD}}(s,a,G) \triangleq d_{\mathrm{QMD}}(s, S'_{(s,a)}) + d_{\mathrm{QMD}}(S'_{(s,a)}, G).$$

**Definition 8** (Quasimetric policy in general stochastic setting). *Given goal-conditioned MDP $\mathcal{M}$ with states $\mathcal{S}$, actions $\mathcal{A}$, and goal-conditioned Kronecker delta reward function $r_g(s) = \delta_{(s,g)}$ and quasimetric over distributions $d_{\mathrm{QMD}}$ (Definition 6), we extend the quasimetric policy to stochastic settings:*

$$\pi_d(a \mid s, G) \in \arg\min_a d_{\mathrm{QMD}}(s,a,G). \tag{19}$$

We can also generalize the planning class to take in states, actions, and state distributions as inputs:

$$\mathbf{plan} \triangleq \{\mathrm{PLAN} : \mathcal{S} \times \mathcal{A} \times \Delta(\mathcal{S}) \mapsto \Delta(\mathcal{S})\}. \tag{20}$$

This planner chooses a waypoint distribution conditioned on a given start state, action taken from this state, and a desired future goal distribution.

**Definition 9** (Quasimetric planner class in general stochastic setting). *Given goal-conditioned MDP $\mathcal{M}$ and quasimetric over distributions $d_{\mathrm{QMD}}$ (Definition 6), we extend the quasimetric planning class to stochastic settings:*

$$\mathbf{plan}_d \triangleq \{\mathrm{PLAN} \in \mathbf{plan} \mid d_{\mathrm{QMD}}(s,a,W) + d_{\mathrm{QMD}}(W,G) = d_{\mathrm{QMD}}(s,a,G)$$
$$\text{for all } (s,a,G) \in \mathcal{S} \times \mathcal{A} \times \Delta(\mathcal{S}) \text{ where } \mathrm{PLAN}(s,a,G) = W\}.$$

The existence of planning invariance in stochastic settings follows from these quasimetric definitions.

**Lemma 4** (Quasimetric policies are planning invariant in general stochastic settings). *Given an MDP with states $\mathcal{S}$, actions $\mathcal{A}$, and goal-conditioned Kronecker delta reward function $r_g(s) = \delta_{(s,g)}$, consider a quasimetric policy $\pi_d(a \mid s, G)$ and planner class $\mathbf{plan}_d$. Then, for every planning operator*

$$\mathrm{PLAN}_d(s,a,G) = W_{\mathrm{PLAN}} \in \arg\min_{W \in \Delta(\mathcal{S})} (d_{\mathrm{QMD}}(s,a,W) + d_{\mathrm{QMD}}(W,G)),$$

*there exists a quasimetric policy $\pi_d(a \mid s, G)$ such that*

$$\pi_d(a \mid s, G) = \pi_d(a \mid s, W_{\mathrm{PLAN}}),$$

*i.e., the policy is invariant to the planning operator.*

*Proof.* For any start-goal distribution $(s,G) \in \mathcal{S} \times \Delta(\mathcal{S})$ pair,

$$\min_a d_{\mathrm{QMD}}(s,a,G) = \min_a d_{\mathrm{QMD}}(s, S'_{(s,a)}) + d_{\mathrm{QMD}}(S'_{(s,a)}, G) \quad \text{(by definition (Definition 6))}$$

$$= \min_a \min_W d_{\mathrm{QMD}}(s, S'_{(s,a)}) + d_{\mathrm{QMD}}(S'_{(s,a)}, W) + d_{\mathrm{QMD}}(W,G) \quad (\triangle\text{-ineq})$$

$$= \min_a \min_W d_{\mathrm{QMD}}(s,a,W) + d_{\mathrm{QMD}}(W,G) \tag{21}$$

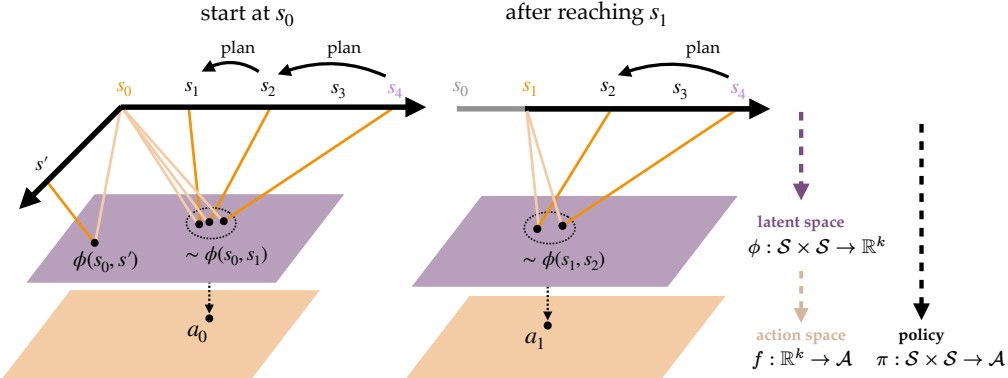

Figure 9: **Invariance to planning leads to horizon generalization.** *(Left)* Invariance to planning maps $(s_0, \{s_1, s_2, s_4\})$ together in latent space, which results in a shared optimal action. *(Right)* After successfully reaching the closest waypoint $s_1$ in 1 step, the next optimal action is also shared, meaning any trajectory of length 2 is optimal. We can repeat this argument for trajectories of length $4, 8, \ldots$ until the entire reachable state space is covered.

From Definition 8, let quasimetric policy $\pi$ be

$$\pi_d(a \mid s, G) \in \arg\min_{a \in \mathcal{A}} d_{\mathrm{QMD}}(s, a, G).$$

Now, applying this policy to state-waypoint distribution pair $(s, W_{\mathrm{PLAN}}) \in \Delta \times \Delta(\mathcal{S})$,

$$
\begin{aligned}
\pi(a|s, W_{\mathrm{PLAN}}) &\in \arg\min_{a \in \mathcal{A}} d_{\mathrm{QMD}}(s, a, W_{\mathrm{PLAN}}) \\
&= \arg\min_{a \in \mathcal{A}} d_{\mathrm{QMD}}(s, a, W_{\mathrm{PLAN}}) + d_{\mathrm{QMD}}(W_{\mathrm{PLAN}}, G) \\
&\subseteq \arg\min_{a \in \mathcal{A}} d_{\mathrm{QMD}}(s, a, G)
\end{aligned}
\tag{22}
$$

as desired. Thus, for any quasimetric planner $\mathrm{PLAN}_d(s, a, G)$, there exists some quasimetric policy

$$
\begin{aligned}
\pi_d(a \mid s, G) = \pi_d(a \mid s, W_{\mathrm{PLAN}}) &\in \arg\min_{a \in \mathcal{A}} d_{\mathrm{QMD}}(s, a, W_{\mathrm{PLAN}}) \\
&\subseteq \arg\min_{a \in \mathcal{A}} d(s, a, G),
\end{aligned}
\tag{23}
$$

as desired. $\qquad\square$

### B.4 Horizon generalization exists

In this section, we include a proof of Theorem 2 for the general stochastic setting. This result is visualized in Fig. 9.

**Theorem 2** (Horizon generalization exists). *Consider an MDP with states $\mathcal{S}$, actions $\mathcal{A}$, and goal-conditioned Kronecker delta reward $r_g(s) = \delta_{(s,g)}$. For any quasimetric $d(s, g)$ over the start-goal space $\mathcal{S} \times \mathcal{S}$ and $c > 0$, the quasimetric policy $\pi_d(a \mid s, g)$ that is optimal over $\mathcal{B}_c = \{(s, g) \in \mathcal{S} \times \mathcal{S} \mid d(s, g) < c\}$ is optimal over the entire start-goal space $\mathcal{S} \times \mathcal{S}$.*

*Proof.* We prove the more general result for policies $\pi_d(a \mid s, G)$ defined over start-goal distribution pairs $(s, G)$. See earlier sections in Appendix B.3 for quasimetric, policy, and planning definitions over distributions.

**Lemma 5.** *For a goal-conditioned MDP with states $\mathcal{S}$, actions $\mathcal{A}$, and goal-conditioned Kronecker delta reward function $r_g(s) = \delta_{(s,g)}$. Let finite thresholds $c > 0$ and quasimetrics $d_{\mathrm{QMD}}(s, G)$ over the start-goal distribution space $\mathcal{S} \times \Delta(\mathcal{S})$ be given. Then, a quasimetric policy $\pi_d(a \mid s, G)$ that is optimal over $\mathcal{B}_c = \{(s, G) \in \mathcal{S} \times \cdot(\mathcal{S}) \mid d(s, G) < c\}$ is optimal over the entire start-goal distribution space $\mathcal{S} \times \Delta(\mathcal{S})$.*

Note that we can recover the fixed, deterministic action and goal setting ("fixed" setting) by letting goal-distribution $G$ be a Dirac delta function at a single goal $g$.

We prove Lemma 5 using induction. Assume optimality over $\mathcal{B}_n = \{(s, G) \in \mathcal{S} \times \Delta(\mathcal{S}) \mid d(s, G) < c2^n\}$ for arbitrary $n \in \mathbb{Z}^+$. Without loss of generality, consider arbitrary state $s \in \mathcal{S}$ and all goal distributions $\mathcal{B}_n(s) = \{G \in \Delta(\mathcal{S}) \mid d(s, G) < c2^n\}$.

We can consider the space of distributions $\mathcal{B}_n(G)$ that are $c2^n$ distance away from goal distribution $G \in \mathcal{B}_n(s)$:

$$\begin{aligned}
\mathcal{B}_n(G) &= \{S' \in \Delta(\mathcal{S}) \mid d(G, S') < c2^n, G \in \mathcal{B}_n(s)\} \\
&= \{S' \in \Delta(\mathcal{S}) \mid d(s, S') < c2^{n+1}\} \\
&= \mathcal{B}_{n+1}(s)
\end{aligned} \tag{24}$$

where we correspondingly define the ball of start-goal distribution pairs drawn from $\mathcal{B}_n(G)$ as

$$\begin{aligned}
\mathcal{B}'_n &= \{(s, S') \in \mathcal{S} \times \Delta(\mathcal{S}) \mid S' \in \mathcal{B}_n(G), G \in \mathcal{B}_n(s)\} \\
&= \{(s, S') \in \mathcal{S} \times \Delta(\mathcal{S}) \mid S' \in \mathcal{B}_{n+1}(s)\} \\
&= \mathcal{B}_{n+1}.
\end{aligned} \tag{25}$$

Invoking the definition of the quasimetric policy $\pi_d(a \mid s, S')$, for some waypoint distribution $W_{\text{PLAN}} \in \arg\min_{W \in \mathcal{B}_{n+1}(s)}(d_{\text{QMD}}(s, a, W) + d_{\text{QMD}}(W, G))$ and goal distribution $G \in \mathcal{B}_{n+1}(s)$:

$$\pi_d(a \mid s, G) \in \arg\min_{a \in \mathcal{A}} d_{\text{QMD}}(s, a, W_{\text{PLAN}}).$$

To show that there always exists some planned waypoint distribution $W_{\text{PLAN}}$ within the region of assumed optimality $\mathcal{B}_n$ from the inductive assumption, we consider the case $W_{\text{PLAN}} \notin \mathcal{B}_n(s)$ and show that there exists some $W_{\text{PLAN, IN}} \in \mathcal{B}_n$ such that

$$d_{\text{QMD}}(s, a, W_{\text{PLAN, IN}}) + d_{\text{QMD}}(W_{\text{PLAN, IN}}, G) = d_{\text{QMD}}(s, a, G).$$

We drop the QMD subscript on quasimetric $d$ for readability. By the triangle inequality,

$$\begin{aligned}
d(s, a, G) &= \min_{W \in \mathcal{B}_{n+1}(s)} (d(s, a, W) + d(W, G)) \\
&= d(s, a, W_{\text{PLAN}}) + d(W_{\text{PLAN}}, G) \\
&= \min_{W_{\text{OUT}} \in \mathcal{B}_{n+1}(s) \backslash \mathcal{B}_n(s)} d(s, a, W_{\text{OUT}}) + d(W_{\text{OUT}}, G) \\
&= \min_{W_{\text{OUT}} \in \mathcal{B}_{n+1}(s) \backslash \mathcal{B}_n(s)} \min_{W_{\text{IN}} \in \mathcal{B}_n} (d(s, a, W_{\text{IN}}) + d(W_{\text{IN}}, W_{\text{OUT}})) + d(W_{\text{OUT}}, G) \\
&= \min_{W_{\text{IN}} \in \mathcal{B}_n(s)} \min_{W_{\text{OUT}} \in \mathcal{B}_{n+1}(s) \backslash \mathcal{B}_n(s)} d(s, a, W_{\text{IN}}) + (d(W_{\text{IN}}, W_{\text{OUT}}) + d(W_{\text{OUT}}, G)) \\
&= \min_{W_{\text{IN}} \in \mathcal{B}_n(s)} d(s, a, W_{\text{IN}}) + d(W_{\text{IN}}, G) \qquad\qquad (\triangle\text{-ineq}) \\
&= d(s, a, W_{\text{PLAN, IN}}) + d(W_{\text{PLAN, IN}}, G),
\end{aligned} \tag{26}$$

for all $s \in \mathcal{S}$. Thus, there always exists an optimal state-waypoint distribution pair within the assumed optimality region $\mathcal{B}_n$; we can thus restrict $(s, W_{\text{PLAN}}) \in \mathcal{B}_n$.

Therefore, with the previously defined quasimetric policy $\pi_d(a \mid s, G)$,

$$\begin{aligned}
\pi_d(a \mid (s, W_{\text{PLAN}}) \in \mathcal{B}_n) &\in \arg\min_{a \in \mathcal{A}} d(s, a, W_{\text{PLAN}}) && \text{(inductive assumption)} \\
&\subseteq \arg\min_{a \in \mathcal{A}} d(s, a, G),, && \text{(Lemma 4: planning invariance)}
\end{aligned}$$

so, the policy $\pi_d(a \mid s, G)$ is optimal over $\mathcal{B}_{n+1}$ following the inductive assumption. Since we assume the base case holds everywhere, Theorem 2 follows. $\qquad\square$

## B.5 HORIZON GENERALIZATION IS NONTRIVIAL

We observe that planning invariance and horizon generalization can be arbitrarily violated for general policies and MDPs.

**Remark 3** (Horizon generalization is nontrivial). *Let finite $c > 0$ and goal-conditioned MDP with states $\mathcal{S}$, actions $\mathcal{A}$, and goal-conditioned Kronecker delta reward function $r_g(s) = \delta_{(s,g)}$ be given where there are no states outside of $\mathcal{S}$. For a policy that is not planning invariant, optimality over $\mathcal{B}_c = \{(s,g) \in \mathcal{S} \times \mathcal{S} \mid d(s,g) < c\}$ is not a sufficient condition for optimality over the entire start-goal space $\mathcal{S} \times \mathcal{S}$.*

*Proof.* We restrict our proof to the fixed, controlled setting and let quasimetric $d(s,g)$ be the successor distance $d_{\mathrm{SD}}(s,g)$ (Myers et al., 2024a) — this assumption lets us directly equate the optimal horizon $H$ to the distance $d_{\mathrm{SD}}(s,g)$, but note that similar arguments can be applied by treating $d(s,g)$ as a generalized notion of horizon.

Consider the goal-conditioned policy $\pi^{*,H}(a \mid s,g)$ that is optimal for $(s,g)$ pairs over some horizon $H$. Assume there is at least one goal $g'$ that is optimally $H + 1$ actions away from $s$, and that there exists some optimal waypoint $s'$ on the way to $g'$ reachable via actions $\mathcal{A}' \subset \mathcal{A}$ (where $\mathcal{A} \setminus \mathcal{A}'$, the set of suboptimal actions, is nonempty).

We can then construct a policy $\pi^{H+1}$ where (1) $\pi^{H+1}(a \mid s,g')$ returns an action in the suboptimal set $\mathcal{A} \setminus \mathcal{A}'$ and (2) $\pi^{H+1}$ restricted to start-goal pairs horizon $H$ apart is equivalent to $\pi^{*,H}$. Therefore, an arbitrary, non-planning invariant goal-reaching policy does not necessarily exhibit horizon generalization. $\square$

## C    Which Practical Methods Might Exhibit Horizon Generalization?

In this section, we discuss how temporal difference methods, quasimetric architectures, RL algorithms, and data augmentations that employ explicit planning can all achieve planning invariance under some assumptions. Appendix C.1 discusses several new directions for designing RL algorithms that are invariant to planning. Appendix C.2 recalls figures from prior works in search of evidence for horizon generalization.

**Dynamic programming and temporal difference (TD) learning.** We expect that dynamic programming and TD methods will achieve planning invariance in tabular settings. The intuition is that TD methods "stitch" (Ziebart et al., 2008) together trajectories which is a natural route to obtain policies with horizon generalization. Indeed, our definition of planning invariance is very closely tied with the optimal substructure property (Cormen et al., 2022, pp. 382-387) of dynamic programming *problems*, and likely could be redefined entirely in terms of optimal substructure. Viewing horizon generalization and planning invariance through the lens of machine learning allows us to consider a broader set of tools for achieving invariance and generalization (e.g., special neural network layers, data augmentation).

**Quasimetric Architectures (implicit planning).** Prior methods that employ special neural networks may have some degree of horizon generalization. For example, some prior methods (Wang et al., 2023; Pitis et al., 2020; Myers et al., 2024a) use quasimetric networks to represent a distance function. As the correct distance function satisfies the triangle inequality, it is useful to employ special quasimetric neural network architectures (Liu et al., 2023; Wang and Isola, 2022b;a) that are guaranteed to satisfy the same property before seeing any training data. These architectures are invariant to path relaxation by construction, though prior work rarely examines their generalization or invariance properties. Other prior methods (Tamar et al., 2016; Lee et al., 2018) have proposed architectures that perform value iteration internally and (hence) may be invariant to the Bellman operator. Because Bellman optimality implies invariance to optimal planning (c.f. optimal substructure), we expect that these value iteration networks to exhibit some horizon generalization as well.

**Explicit planning methods.** While our proof of planning used a specific notion of planning, prior work has proposed RL methods that employ many different styles of planning: graph search methods (Savinov et al., 2018; Zhang et al., 2021b; Beker et al., 2022; Chane-Sane et al., 2021), model-based methods (Sutton, 1991; Chua et al., 2018; Nagabandi et al., 2018; Lowrey et al., 2018; Williams et al., 2017), collocation methods (Rybkin et al., 2021), and hierarchical methods (Kulkarni et al., 2016; Parascandolo et al., 2020; Nasiriany et al., 2019; Pertsch et al., 2020). Insofar as these methods approximate the method used in our proof, it is reasonable to expect that they may achieve some degree of planning invariance and

horizon generalization (see Fig. 10). Prior methods in this space are typically evaluated on the *training* distribution, so their horizon generalization capabilities are typically not evaluated. However, the improved generalization properties might have still contributed to the faster learning on the *training* tasks: after just learning the easier tasks, these methods would have already solved the complex tasks, leading to higher average success rates.

**Data augmentation.** Finally, prior work (Ghugare et al., 2024; Chane-Sane et al., 2021) has argued that data augmentation provides another avenue for achieving the benefits typically associated with planning or dynamic programming.

## C.1   NEW METHODS FOR PLANNING INVARIANCE

While the aim of this paper is not to propose a new method, we will discuss several new directions that may be examined for achieving planning invariance.

**Representation learning.** As shown in Fig. 2, planning invariance implies that some internal representation inside a policy must map start-goal inputs and start-waypoint inputs to similar representations. What representation learning objective would result in representations that, when used for a policy, guarantee horizon generalization?[2] The fact that plans over representations sometimes correspond to geodesics (Tenenbaum et al., 2000; Eysenbach et al., 2024) hints that this may be possible.

**Flattening hierarchical methods.** While hierarchical methods often achieve higher success rates in practice, it remains unclear why flat methods cannot achieve similar performance given the same data. While prior work has suggested that hierarchies may aid in exploration (Nachum et al., 2019), it may be the case that they (somehow) exploit the metric structure of the problem. Once this inductive bias is identified, it may be possible to imbue it into a "flat" policy so that it can achieve similar performance (without the complexity of hierarchical methods).

**Policies that learn to plan.** While explicit planning methods may be invariant to planning, recent work has suggested that certain policies can *learn* to plan when trained on sufficient data (Chane-Sane et al., 2021; Lee et al., 2024). Insofar as neural networks are universal function approximators, they may learn to approximate a planning operator internally. The best way of learning such networks that implicitly learn to perform planning remains an open question.

## C.2   EVIDENCE OF HORIZON GENERALIZATION AND PLANNING INVARIANCE FROM PRIOR WORK

Not only do the experiments in Section 6 provide evidence for horizon generalization and planning invariance, but we also can find evidence of these properties in the experiments run by prior work. This section reviews three such examples, with the corresponding figures from prior work in Fig. 10:

1. Zhang et al. (2021b) propose a method for goal-conditioned RL in the online setting that performs planning during exploration. While not the main focus of the paper, an ablation experiment in that paper hints that their method may have some degree of planning invariance: after training, the policy produced by their method is evaluated both with and without planning, and achieves similar success rates. This experiment hints at another avenue for achieving planning invariance: rather than changing the architecture or learning rule, simply changing how data are collected may be sufficient.

2. Ghugare et al. (2024) propose a method for goal-conditioned RL in the offline setting that performs temporal data augmentation. Their key result, reproduced above, is that the resulting method generalizes better to unseen start-goal pairs, as compared with a baseline. While this notion of generalization is not exactly the same as horizon generalization (unseen start-goal pairs may still be close to one another), the high success rates of the proposed method suggest that method does not *just* generalize to nearby start-goal pairs, but also exhibits horizon generalization by succeeding in reaching unseen distant start-goal pairs.

---

[2]The construction in our proof is a degenerate case of this, where the internal representations are equal to the output actions.

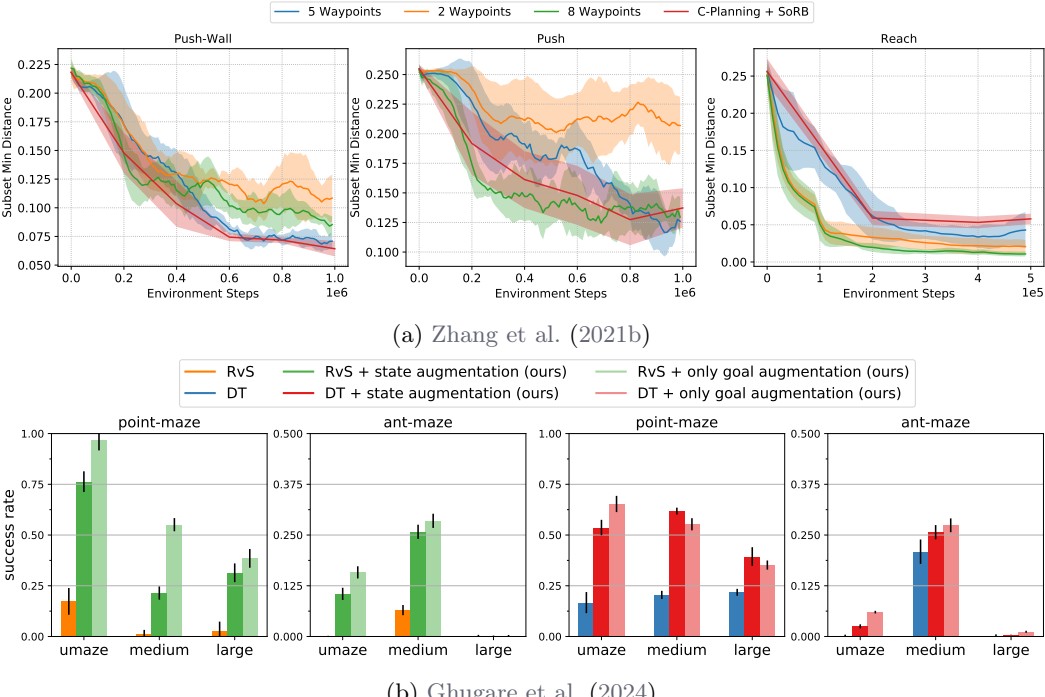

(a) Zhang et al. (2021b)

(b) Ghugare et al. (2024)

Figure 10: **Evidence of Horizon Generalization and Planning Invariance from Prior work.** *(a)* Prior work has observed that if policies are trained in an online setting and perform planning during exploration, then those policies see little benefit from doing planning during evaluation. This observation suggests that these policies may have learned to be planning invariant. While results are not stratified into training and testing tasks, we speculate that the faster learning of that method (relative to baselines, not shown) may be explained by the policy generalizing from easy tasks (which are learned more quickly) to more difficult tasks. *(b)* Prior work studies how data augmentation can facilitate combinatorial generalization. While the notion of combinatorial generalization studied there is slightly from horizon generalization, a method that performs combinatorial generalization would also achieve effective horizon generalization.

## D    SELF-CONSISTENT MODELS

In machine learning, we usually strive for *consistent* models: ones that faithfully predict the training data. Sometimes (often), however, a model that is consistent with the training data may be inconsistent with other yet-to-be-seen training examples. In the absence of infinite data, one way of performing model selection is to see whether a model's predictions are self-consistent with one another. This is perhaps most easily seen in the case of metric learning, as studied in this paper. If we are trying to learn a metric $d(x, y)$, then the properties of metrics tell us something about the predictions that our model should make, both on seen and unseen inputs. For example, even on unseen inputs, our model's predictions should obey the triangle inequality. Given many candidate models that are all consistent with the training data, we may be able to rule out some of those models if their predictions on unseen examples are not "logically" consistent (e.g., if they violate the triangle inequality). *One way of interpreting quasimetric neural networks is that they are architecturally constrained to be self-consistent.* We will discuss a few implications of this observation.

**Do self-consistent models know what they know?** What if we assume that quasimetric networks can generalize? That is, after learning that (say) $s_1$ and $s_2$ are 5 steps apart, it will predict that similar states $s_1'$ and $s_2'$ are also 5 steps apart. Because the model is architecturally constrained to be a quasimetric, this prediction (or "hallucination") could also result in changing the predictions for other $s$-$g$ pairs. That is, this new "hallucinated" edge $s_1' \longrightarrow s_2'$ might result in path relaxation for yet other edges.

**What other sorts of models are self-consistent?** There has been much discussion of self-consistency in the language-modeling literature (Huang et al., 2022; Irving et al., 2018). Many of these methods are predicated on the same underlying as self-consistency in quasimetric networks: checking whether the model makes logically consistent predictions on unseen inputs. Logical consistency might be used to determine that a prediction is unlikely, and so the model can be updated or revised to make a different prediction instead.

There is an important difference between this example and the quasimetrics. While the axiom used for checking self-consistency in quasimetrics was the triangle inequality, in this language modeling example self-consistency is checked using the predictions from the language model itself. In the example of quasimetrics, our ability to precisely write down a mathematical notion of consistency enabled us to translate that axiom into an architecture that is self-consistent with this property. This raises an intriguing question: *Can we quantify the rules of logic in such a way that they can be translated into a logically self-consistent language model?* What makes this claim seem alluringly tangible is that there is abundant literature from mathematics and philosophy on quantifying logical rules (Whitehead and Russell, 1927).

**MDP reductions.** Finally, is it possible to map one MDP to another MDP (e.g., with different observations, with different actions) so that any RL algorithm applied to this transformed MDP automatically achieves the planning invariance property?

## E    EXPERIMENT DETAILS

Table 1: Summary of methods and modifications tested

| Method | Description | | Losses | Critics |
|--------|-------------|---|--------|---------|
| CRL | Contrastive RL (Eysenbach et al., 2022) | | $\{\mathcal{L}_{\text{fwd}}, \mathcal{L}_{\text{bwd}}, \mathcal{L}_{\text{sym}}\}$ | $\{d_{\ell_2}, d_{\text{MLP}}\}$ |
| SAC | Soft Actor-Critic (Haarnoja et al., 2018) | | $\{\mathcal{L}_{\text{sac}}\}$ | $\{Q_{\text{MLP}}\}$ |
| CMD-1 | Contrastive metric distillation (Myers et al., 2024a) | | $\{\mathcal{L}_{\text{bwd}}\}$ | $\{d_{\text{MRN}}\}$ |

| (a) Losses | | (b) Architectures | |
|---|---|---|---|
| $\mathcal{L}_{\text{fwd}}$ | InfoNCE loss: predict goal $g$ from current state-action $(s, a)$ pair (Sohn, 2016) | $d_{\ell_2}$ | L2-distance parameterized architecture, uses $\|\phi(s) - \psi(g)\|$ as a distance/critic (Eysenbach et al., 2024) |
| $\mathcal{L}_{\text{bwd}}$ | Backward InfoNCE loss: predict current state and action $(s, a)$ from future state $g$ (Bortkiewicz et al., 2024) | $d_{\text{MLP}}$ | Uses multi-layer perceptron (MLP) to parameterize the distance/critic (Rosenblatt, 1961; Burr, 1986) |
| $\mathcal{L}_{\text{sym}}$ | Symmetric contrastive loss: combine the forward and backward contrastive losses (Radford et al., 2021) | $d_{\text{MRN}}$ | Metric residual network, uses a quasimetric architecture to parameterize the distance/critic (Liu et al., 2023) |
| $\mathcal{L}_{\text{sac}}$ | Temporal difference loss (Haarnoja et al., 2018) | $Q_{\text{MLP}}$ | MLP-parameterized Q-function (Haarnoja et al., 2018) |

The following subsections discuss the environment details for the figures in the main text.

### E.1    DIDACTIC MAZE: FIGURE 2

This task is a maze with walls shown as in Fig. 2. The dynamics are deterministic. There are 5 actions, corresponding to the cardinal directions and a no-op action.

For this plot, we generated data from a random policy, using 1000 trajectories of length 200. We estimated distances using Monte Carlo regression. The left two subplots were generated by selecting actions uses these Monte Carlo distances. We computed the true distances by running Dijkstra's algorithm. The right two subplots show actions selected using Dijkstra's algorithm.

### E.2 Tabular Maze Navigation: Figure 4 (Top)

This plot used the same environment as described in Appendix E.1. For this plot, we generated 3000 trajectories of length 50 using a random policy. Only 14% of start-goal pairs have any trajectory between them, meaning that the vast majority of start-goal pairs have never been seen together during training. Thus, this is a good setting for studying generalization.

We first estimated distances using Monte Carlo regression. We select actions using a Boltzmann policy with temperature 0.1 (i.e., $\pi(a \mid s, g) \propto e^{-0.1d(s,g)}$). Evaluation is done over 1000 randomly-sampled start-goal pairs. The X axis is binned based on the shortest path distance. The data are aggregated so that start-goal pairs with distance between (say) 20 and 30 get plotted at $x = 30$. The "metric regression + quasimetric" distances are obtained by performing path relaxation on these Monte Carlo distances until convergence. The corresponding policy is again a Boltzmann policy with temperature 0.1.

For the Top Right subplot, we perform planning using Dijkstra's algorithm. We first identify a set of candidate midpoint states where $d(s, w)$ and $d(w, g)$ are both within one unit of half the shortest path distance. We then randomly sample a midpoint state. This planning is done anew at every timestep.

### E.3 Learned Maze Navigation: Figure 4 (Bottom)

This plot used the same environment as described in Appendix E.1. The CRL method refers to (Eysenbach et al., 2022) and CMD refers to (Myers et al., 2024a). We used a representation dimension of 16, a batch size of 256, neural networks with 2 hidden layers of width 32 and Swish activations, $\gamma = 0.9$, and Adam optimizer with learning rate $3 \cdot 10^{-3}$. The loss functions and architectures are based on those from (Bortkiewicz et al., 2024).

For the Bottom Right subplot, we performed planning in the same way as for the Top Right subplot.

### E.4 JaxGCRL Benchmark Environments

**Ant (Figure 5):** For this task we used a version of the Ant environment from Bortkiewicz et al. (2024) modified to have variable start positions and distances to the goal. All other hyperparameters are kept as the defaults from that paper. Training is done for 100M steps

**AntMaze and Humanoid (Figure 6):** Both environments are modified versions of the AntMaze and Humanoid environments from Bortkiewicz et al. (2024). The CMD-1 and CRL methods (with the backward infoNCE loss and $\ell_2$ distance parameterization) were evaluated at the listed distances and twice as far, and the ratio of the two success rates was used to compute $\eta$.

### E.5 Long Maze: Figure 8

For this experiment we used an S-shaped maze, shown in Fig. 11.[3] The dynamics are the same as those of Fig. 2.

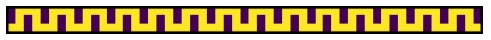

Figure 11: Long S-shaped maze.

We collected 3000 trajectories of length 10 and applied CRL with a representation dimension of 16, a batch size of 256, neural networks with 2 hidden layers of width 32 and Swish activations, the backward NCE loss (Bortkiewicz et al., 2024), $\gamma = 0.9$, using the Adam optimizer with learning rate $3 \cdot 10^{-3}$. We measured the Bellman error as follows, where $x_0, x_1, x_T$ are the current, immediate next, and future states:

```
1  pdist = metric_fn.apply(params, x0[:, None], xT[None])
2  pdist_next = metric_fn.apply(params, x1[:, None], xT[None])
3  td_target = (1 - gamma) * (x1 == xT[None, :, 0])
4          + gamma * jax.nn.softmax(pdist_next, axis=1)
5  bellman = optax.kl_divergence(
```

---

[3]We used this maze in preliminary versions of other experiments, but opted for the larger maze in the other paper experiments because the results were easier to visualize.

```
6            td_target, jax.nn.softmax(pdist, axis=1)
7   ).mean()
```

For the success rates in the Left subplot, we stratify goals into "easy" (less than 100 steps away, under an optimal policy) and "distant" (more than 100 steps away).

We repeated this experiment 10 times for generate the standard errors shown in both the Left and Right subplots.

