# OpenReview forum: "Horizon Generalization in Reinforcement Learning"
_ICLR.cc/2025/Conference — ICLR 2025 Poster_

### Official Review · Reviewer_6v2T · 2024-10-22

**Soundness:** 3
**Presentation:** 3
**Contribution:** 3
**Rating:** 6
**Confidence:** 2

**Summary:**

The motivation and insight presented in this paper are interesting and straightforward. This paper presents the abundant theoretical contribution to Planning Invariance and Horizon Generalization. Experimental results of existing methods support the theoretical results.

**Strengths:**

1. Good writing and structure.
2. The authors illustrate their research problem via the good toy example (Figrue 3).
3. Theoretical results are clear and sound.

**Weaknesses:**

I have no concern with the main contribution of the theoretical part of this paper. The main concern is related to the application and limitations.

1. One of the main concerns of the reviewer is the limitations of the planning invariance and horizon generalization, as mentioned by the authors (Section 4.5).
2. I am also concerned about real-world applications (e.g., more complicated tasks). In real-world settings, the planning invariance and horizon generalization seem to be limited in their application to general cases, although they indeed exist as mentioned by the authors. In the high-dimensional settings (Section 6.2), it is hard to derive insightful results.
3. The authors provide some interesting future directions in Appendix C, but it would be better to present some experimental results or practical implementations.

**Questions:**

See weakness.

---

> ### Author Response · Authors · 2024-11-22
> **Rebuttal**
>
> Dear reviewer,
>
> Thank you for your responses and suggestions for improving the paper.
>
> We have revised the text (changes highlighted in red) to improve the clarity of our theoretical claims, and clearly state the practical implications of our theory and experiments for GCRL algorithms throughout the paper.
> Regarding your concern about real-world applications and limitations of these generalizations in more complex settings, we will evaluate a higher dimensional setting (humanoid) with a more challenging control component. We hope that these experiments will sufficiently address any remaining concerns.
>
> > I am also concerned about real-world applications (e.g., more complicated tasks). In real-world settings, the planning invariance and horizon generalization seem to be limited in their application to general cases, although they indeed exist as mentioned by the authors. In the high-dimensional settings (Section 6.2), it is hard to derive insightful results.
>
> As mentioned in Section 4.5, (1) the optimality over local trajectories and the (2) coverage of local trajectories are both concerns. For (1), there are unavoidable sources of errors from function approximation and inherent noise in training. To push on this potential weakness, **we will further test the presence and relationship between planning invariance and horizon generalization settings in a higher d.o.f. settings like Humanoid [1]**.
>
> We would also like to highlight that we have already tested planning invariance, horizon generalization, and their relationship in Ant, which is a standard, 27-dim observation 8DoF control, RL benchmark used as a primary evaluation in many recent prior RL work [3,4,5]. Since the initial submission, **we have extended the results in [Figure 5](https://gcdnb.pbrd.co/images/Gv5HtRmtwnhR.png) to show further horizon generalization in this setting.**
>
> ### References
>
> [1] Tunyasuvunakool, S. et al., 2020. "dm_control: Software and Tasks for Continuous Control." *Software Impacts*
>
> [2] Park, S. et al., 2024. "OGBench: Benchmarking Offline Goal-Conditioned RL." arXiv:2410.20092
>
> [3] Zheng, C. et al., 2023. "Contrastive Difference Predictive Coding." *ICLR*
>
> [4] Fujimoto, S. et al., 2018. "Addressing Function Approximation Error in Actor-Critic Methods." *ICML*
>
> [5] Freeman, C. et al., 2021. "Brax—a Differentiable Physics Engine for Large Scale Rigid Body Simulation." *NeurIPS Datasets and Benchmarks*

---

> > ### Author Response · Authors · 2024-11-25
> > **Rebuttal follow-up**
> >
> > Dear Reviewer,
> >
> > We have worked hard to incorporate the review feedback by running new experiments and revising the paper. We'd really appreciate if you could confirm whether these changes address the concerns about the paper. If we have misunderstood any of the concerns, we'd like to learn that now so that we can further revise the paper or run additional experiments.
> >
> > Thanks!
> >
> > The Authors

---

> > > ### Author Response · Authors · 2024-11-30
> > > **Rebuttal follow-up**
> > >
> > > Dear Reviewer,
> > >
> > > We have worked hard to incorporate the review feedback by running new experiments and revising the paper. **Do the revisions and discussions above address your concerns?** We would greatly appreciate your engagement.
> > >
> > > Thanks!
> > >
> > > The Authors

---

### Official Review · Reviewer_ctUj · 2024-11-02

**Soundness:** 2
**Presentation:** 2
**Contribution:** 2
**Rating:** 5
**Confidence:** 3

**Summary:**

This paper studies horizon generalization in goal-conditioned RL. The main contributions of the paper are a definition of horizon generalization and planning invariance, and proofs of existence for both, as well as some small-scale experimental studies. I admit I am unclear what the takeaways for the reader are - it seems like the main contributions, i.e. the definition of horizon and planning invariance, are quite obvious, as well as the existence of these kinds of invariances. I do agree that the goal of inducing invariances that facilitate generalization in RL is important. It may be that I misunderstood parts of the paper, hence I am putting my confidence at 3 and recommendation at borderline.

**Strengths:**

- Conceptual investigations are welcome
- The paper's figures are well drawn

**Weaknesses:**

- As mentioned above, the takeaways for the reader are not clear - this does not propose a new method, or (to my understanding) clearly shed light on existing methods, highlight a previously unknown weakness,...
- The relationship of this work to other areas of work is unclear in some cases.

Please see my detailed questions/comments below.

**Questions:**

- It seems obvious that if the series of waypoints produced by a planner is optimal, then we should have planning invariance according to the definition given here (i.e., the optimal action to reach the next waypoint is the same at the optimal action to reach the final goal), so my guess is that this is not the main message of the paper. The other way around, where the agent trains on easy goals and generalizes to hard ones, seems much more useful, so that this is what the paper is getting at. The paper refers to this as the “forward perspective”. However, there doesn't seem to be any suggestions on how to induce this kind of invariance. Also, this seems related to compositional generalization in language-conditioned RL, where the agent is trained on language commands and asked to generalize to novel combinations. How does this relate to planning invariance/generalization?

- The paper claims that most prior work on generalization in RL relates to invariance to perceptual changes or simulation parameters. But there is a rich literature on state abstractions which induce invariances to criteria other than perceptual differences, for example: optimal actions (known as policy irrelevance), rewards/Q-values (utile distinctions) [1] bisimulation [2]  belonging to a common latent state (however defined) [3]. It seems like the proposed planning invariance could be seen as a special case of one of these other abstractions (such as policy irrelevance). It would be helpful to include a discussion of other types of state abstraction/generalization in the paper. If the proposed invariance cannot be subsumed in any of the prior ones, it would be helpful to describe in what ways.

- It would help to give more examples of non-trivial cases where planning invariance is or is not met, in order to build intuitions. There is one example in Section 4.1, but it didn’t really help my understanding because in that example, there was a single action that was optimal for all goals (and the problem is in a sense trivial for that reason). Also, the example of knocking over dominoes with levers felt fairly contrived. Would it be possible to add one or more examples that 1) have different actions that are optimal for different goals, 2) one can construct some policies that do have planning invariance and others not, in order to illustrate their differences, and 3) are more closely related to common sequential decision-making problems? I know there is also the example in Figure 6, but there the goal-conditioned policy on the left is not optimal.

[1] https://citeseerx.ist.psu.edu/document?repid=rep1&type=pdf&doi=ca9a2d326b9de48c095a6cb5912e1990d2c5ab46

[2] https://www.auai.org/uai2014/proceedings/individuals/67.pdf

[3] https://arxiv.org/pdf/1901.09018

---

> ### Author Response · Authors · 2024-11-22
> **Rebuttal**
>
> Dear Reviewer,
>
> Thank you for your responses and suggestions.
>
> It seems your main suggestion is to provide more intuition for the theoretical claims, including a discussion of the relationship with prior work on representations in RL. We have made revisions in the paper to address these concerns and believe that the paper motivation is now much clearer. **Together with the discussion below, do these fully address your concerns?**
>
> We have revised to the paper to provide more intuition and background:
> 1. **Highlight relationships with prior state abstraction works such as bisimulation**: In Section 2, we now compare and contrast horizon generalization with other forms of generalization captured by prior state abstraction methods. We highlight that, to our knowledge, there has not been prior work directly addressing generalization from short to long horizons. Furthermore, planning invariant policies map state-action-waypoint pairs and state-action-goal pairs to the same latents, despite different associated rewards and Q-values (see Section 3 for relation between quasimetric and underlying reward function in an MDP). This is already a violation of the Q-irrelevance relation and the stronger bisimulation relation [1]: the invariances captured by planning invariance and such state abstractions are fundamentally different.
> 2. **Building intuition for planning invariance**: We have updated our planning invariance figure to help build intuition and highlight Section 5: Methods for Planning Invariance: Old and New.
> 3. **Takeaways**: We have highlighted the high-level takeaways for the reader throughout the paper (see end of Section 1, Section 2, end of Section 4.4, and Section 7). By theoretically and empirically linking planning with horizon generalization, our work suggests practical ways (i.e., quasimetric methods) to achieve powerful notions of generalization from short to long horizons.
> **Do these revisions, together with the discussion below, fully address your concerns?** We look forward to continuing the discussion!
>
> > There doesn't seem to be any suggestions on how to induce this kind of invariance.
>
> This invariance is precisely the structure imposed by quasimetric architectures—we have clarified the notation so this is now clearly stated as [Theorem 2](https://gcdnb.pbrd.co/images/gEMhIRNV1hGk.png) in the text.
> We have also added a short summary of the concrete takeaways to the conclusion section, and a summary of different architectures and losses and how they relate to the planning invariance of quasimetric architectures.
> Additional discussion is presented Section 5: “Methods for Planning Invariance, Old and New” as well as Appendix C. For example, for the forward perspective, we can induce planning invariance (1) explicitly via dynamic programming or (2) implicitly using a quasimetric architecture. We empirically demonstrate that quasimetric architectures, when combined with a contrastive metric learning approach [3], are relatively planning invariant compared to other architectures (see [Figure 5, right](https://gcdnb.pbrd.co/images/Gv5HtRmtwnhR.png)).
>
> > relation to compositional generalization in language-conditioned RL
>
> Indeed, a key advantage of language as a form of task specification is its ability to easily express compositional relationships. Future work could investigate how planning-invariant reasoning could be extended to these alternate task modalities, through, e.g., learned mappings from the task space into the quasimetric goal space [2].
> We have added discussion to our conclusion (marked in red, under “Limitations and future work”) on how future work could tackle these challenges.
>
> > give more examples of non-trivial cases where planning invariance is or is not met, in order to build intuitions
>
> We have replaced the figure in Section 4.1 with [Figure 2](https://gcdnb.pbrd.co/images/PNHteRUKMkhP.png). With this revised figure, we can (1) visualize different 2D translation actions for different navigation goals (purple star and brown star) as well as optimal (tan) and suboptimal (gray) trajectories, (2) identify that the left policy has no planning invariance (policies towards goal and waypoint are not the same, where the policy selects a *suboptimal* action while navigating towards the goal and *optimal* action while navigating towards the waypoint) while the right policy has planning invariance (indicated by the same, optimal policy whether directed towards a goal or the waypoint), and (3) concretely connect the presented maze task to actual sequential decision-making problems like navigation with obstacles.

---

> > ### Author Response · Authors · 2024-11-22
> > **Rebuttal (cont.)**
> >
> > ### References
> >
> > [1] Zhang, A., McAllister, R. et al., 2021. “Learning Invariant Representations for Reinforcement Learning Without Reconstruction.” *ICLR*
> >
> > [2] Jang, E. et al., 2021. "BC-Z: Zero-Shot Task Generalization With Robotic Imitation Learning." *CoRL*
> >
> > [3] Myers, V. et al., 2024. "Learning Temporal Distances: Contrastive Successor Features Can Provide a Metric Structure for Decision-Making." *ICML*

---

> > > ### Author Response · Authors · 2024-11-25
> > > **Rebuttal follow-up**
> > >
> > > Dear Reviewer,
> > >
> > > We have worked hard to incorporate the review feedback by running new experiments and revising the paper. We'd really appreciate if you could confirm whether these changes address the concerns about the paper. If we have misunderstood any of the concerns, we'd like to learn that now so that we can further revise the paper or run additional experiments.
> > >
> > > Thanks!
> > >
> > > The Authors

---

> > > > ### Author Response · Authors · 2024-11-30
> > > > **Rebuttal follow-up**
> > > >
> > > > Dear Reviewer,
> > > >
> > > > We have worked hard to incorporate the review feedback by running new experiments and revising the paper. **Do the revisions and discussions above address your concerns?** We would greatly appreciate your engagement.
> > > >
> > > > Thanks!
> > > >
> > > > The Authors

---

> > > > > ### Comment · Reviewer_ctUj · 2024-12-01
> > > > >
> > > > > Thank you for the response. One thing that I am not clear on is how the definition of planning invariance is not vacuous in many cases, provided the planner is optimal. For example, let's consider a shortest path from start state $s_0$ to goal state $g$. Any planner operating at a coarser resolution must produce waypoints along this path. Then for any state along this path, we will have $\pi(a|s, g) = \pi(a|s, w)$, where $w$ is the next waypoint. So this (optimal) policy must also be planning-invariant. Is it the case that any optimal policy within some region of the state space, must also be planning-invariant within that region? It could be helpful to clarify the relationships between planner optimality, policy optimality, and planning invariance.

---

> > > > > > ### Author Response · Authors · 2024-12-01
> > > > > > **Response**
> > > > > >
> > > > > > > Thank you for the response. One thing that I am not clear on is how the definition of planning invariance is not vacuous in many cases, provided the planner is optimal. For example, let's consider a shortest path from start state $s\_0$ to goal state $g$. Any planner operating at a coarser resolution must produce waypoints along this path. Then for any state along this path, we will have $\pi(a|s, g) = \pi(a|s, w)$, where $w$ is the next waypoint. So this (optimal) policy must also be planning-invariant. Is it the case that any optimal policy within some region of the state space, must also be planning-invariant within that region? It could be helpful to clarify the relationships between planner optimality, policy optimality, and planning invariance.
> > > > > >
> > > > > > Thank you for your suggestions and questions. We would like to clarify the relationships between planner optimality, policy optimality, and planning invariance. Namely, **planning invariance under an optimal planner is a necessary but not sufficient condition for policy optimality, and is a useful inductive bias for a globally optimal policy.** We will make these definitions and relationships clearer in our manuscript. Does the following discussion fully address the reviewer's concerns?
> > > > > >
> > > > > > > Is it the case that any optimal policy within some region of the state space, must also be planning-invariant within that region?
> > > > > >
> > > > > > Planning invariance is defined with respect to a specific planning operator. If a policy is optimal, then the policy is invariant under the optimal planner, but not necessarily invariant under any arbitrary planner. The inverse is not true: a policy that is planning invariant, even under an optimal planner, is not necessarily optimal. Counterexample: we can construct a policy that outputs the same suboptimal actions towards a goal and any optimal waypoint along the shortest path. Thus, planning invariance is an **inductive bias** for a globally optimal policy that can be conveniently enforced via a quasimetric. This view of invariance as an inductive bias is in the same vein as prior ML work enforcing invariances under spatial operators (rotations, translations, etc.) for, e.g., facial recognition tasks [1,2,3]. **Our contribution is to show how ''invariance to planning'' is the analogous invariance property for horizon generalization in RL, and how, like spatial invariances, it can be enforced via architectural constraints.**
> > > > > >
> > > > > > > relationships between planner optimality, policy optimality, and planning invariance
> > > > > >
> > > > > > What makes planning invariance a *useful* inductive bias is that planning invariant policies propagate *local* policy optimality to global optimality. Local policy optimality alone cannot lead to global policy optimality (horizon generalization). However, local policy optimality *combined with* planning invariance under an optimal planner (which is a weaker condition than global policy optimality, see above) leads to global policy optimality. This horizon generalization is nontrivial as we show theoretically (see Remark 3) and empirically (see updated [Fig. 5, right](https://gcdnb.pbrd.co/images/Gv5HtRmtwnhR.png)).
> > > > > >
> > > > > > ### References
> > > > > >
> > > > > > [1] Cohen, T., Welling, M., 2016. ''Group Equivariant Convolutional Networks.'' *ICML*
> > > > > >
> > > > > > [2] Benton, G. et al., 2020. ''Learning Invariances in Neural Networks.'' *NeurIPS*
> > > > > >
> > > > > > [3] Rowley, H., Baluja, S., Kanade, T., 1998. ''Rotation Invariant Neural Network-Based Facial Recognition.'' *IEEE*

---

### Official Review · Reviewer_h6Gn · 2024-11-03

**Soundness:** 1
**Presentation:** 3
**Contribution:** 3
**Rating:** 3
**Confidence:** 4

**Summary:**

This paper explores the problem of horizon generalization in goal-conditioned reinforcement learning. It presents clear definitions of planning invariance and demonstrates how this concept can facilitate horizon generalization.

**Strengths:**

The problem studied in the paper is novel and very interesting to me. The motivation of the paper is clear and the writing of the paper is generally clear and easy to fllow.

The paper presents clear definition of planning invariance and horizon generalization, which make sense to me.

**Weaknesses:**

The paper contains some symbols that are not well defined, some of which even affect the readability of the paper.

- I got lost from Line 245 with $d(s,a,g)$. I don't understand the meaning of this symbol and why it contains an action. The paper didn't clearly define it. This is critical as I failed to fully understand the proof of Lemma 1 and Lemma 2, which are the major results of the paper. I hope I didn't miss anything.

Some undefined symbols that do not affect reading:

- Eq. (2): $p_\gamma^\pi$, $Geom$ is never defined
- Line 117: $\delta$, $r_g$

The paper also uses different terminology from the common terminology, making it seem less professional.

Usually, the distance (metric) function should satisfy the three rules: Non-negativity, Triangle Inequality, and Symmetry; while a quasimetric relaxes one or more of the metric's defining properties.

Minor Comments:

- It seems Lemma 1 and 2 are some important results of the paper. Then please use "Theorem" rather than "Lemma".
- "Markove process" should be "Markov Decision Process". You didn't include the definition of the reward function.



Another weakness is the paper's lack of clarity on how its discoveries could inspire future algorithm development. I hope future revisions of the paper will devote more effort to exploring this aspect.

**Questions:**

I was unable to assess the correctness of the theorems due to some undefined symbols mentioned earlier. Please respond to my previous queries in weakness. I will reassess them in case I missed anything.

---

> ### Author Response · Authors · 2024-11-22
> **Rebuttal**
>
> Dear Reviewer,
>
> Thank you for your responses and suggestions. It seems like your main concerns have to do with notation and clarity, as well as how the theory could relate to practical new methods. We have made revisions in the paper to address these concerns and believe that the paper clarity is now much higher. We hope that these improvements have increased the readability of the main theorems and make the takeaways clear.  **Together with the discussion below, does this fully address your concerns?** We look forward to continuing the discussion!
>
> > Clarifying notation.
>
> We have made several revisions to the paper to improve clarity:
>
> 1. We clearly define the successor distance over actions $d(s,a,g)$, which, using the setup in [1], is $\min\_{\pi} \frac{p^{\pi}\_{\gamma}(s\_K = g|s\_{0} = g)}{p^{\pi}\_{\gamma}(s\_K = g|s\_{0} = s,a)}$ (Equation 5) where $p^{\pi}\_{\gamma}(s\_K = g|s\_{0} = s)$ is the discounted state-occupancy measure (Equation 4) and $p^{\pi}\_{\gamma}(s\_K = g|s\_{0} = s, a)$ is the discounted state-occupancy measure with actions (Equation 5). This is the temporal distance to the goal $g$ conditioned on starting at state $s$ and taking action $a$ under an optimal policy.
> 2. We note that the minimization of $d(s,a,g)$ over actions $a$ is used for action selection: Myers et al. [1] show that $\text{arg}\min_a d(s,a,g)$ corresponds to $\text{arg} \max\_a Q(s,a,g)$ of the Q-function with respect to the goal-conditioned RL MDP.
> 3. We highlight in Sections 3 and 4.2 that the reward for the MDP using the successor distance is $r(s) = \delta\_{(s,g)}$, which is a Kronecker delta function that evaluates to 1 at the goal and 0 at intermediate states.
> 4. We have revised notation in Section 4 and Appendix C to use more standard math terms. For example, we now use set notation to construct $\text{arg} \min$ rather than introduce a “symmetry-breaking” $\text{arg} \min$ over equally optimal actions and waypoints. We have also renamed certain objects for clarity (i.e. policy $\pi^f\Rightarrow \pi^{\text{Fix}}$ and planning operator $\text{Plan}^{f} \Rightarrow \text{Plan}^{\text{Fix}}$ for the deterministic, “fixed” goal, controlled setting).
>
> The changes to the proofs for clarity have been highlighted in red. **Please let us know if additional clarifications will be helpful.**
>
> > How does the paper inspire future algorithm development?
>
> We have added a new discussion in Appendix D about how our theoretical results might inspire future algorithm development, as well as a summary of the concrete takeaways in the conclusion, and a comparison of existing GCRL critic parameterizations and how they relate to quasimetric architectures ([Table 1](https://gcdnb.pbrd.co/images/3LYtfgmtGjGl.png)).
> We agree that the primary takeaway is not a new method, but rather (1) concrete definitions of planning invariance and horizon generalization, (2) proofs linking quasimetrics, planning invariance, and horizon generalization, and (3) experiments showing that horizon generalization is feasible and nontrivial.
> That said, our experiments show that the modified CMD algorithm [1] combined with the backward infoNCE [4] loss provides a good base algorithm that empirically shows some of these properties while being motivated by our theory.
> We believe these results are significant and in line with prior ICLR works that highlight a phenomenon/property of existing methods [2, 3].
>
> > Typos, use of “lemma”
>
> Thank you for these suggestions. We have fixed the typos and added definitions for the terms you noted, and additionally renamed the core (theoretical) results as “theorems” instead of “lemmas.”
>
> ### References
>
> [1] Myers, V. et al., 2024. "Learning Temporal Distances: Contrastive Successor Features Can Provide a Metric Structure for Decision-Making." *ICML*
>
> [2] Richens, J. et al., 2024. "Robust Agents Learn Causal World Models." *ICLR*
>
> [3] Subramani, R. et al., 2024. "On the Expressivity of Objective-Specification Formalisms in Reinforcement Learning." *ICLR*
>
> [4] Bortkiewicz, M. et al., 2024. "Accelerating Goal-Conditioned RL Algorithms and Research." arXiv:2408.11052

---

> > ### Author Response · Authors · 2024-11-25
> > **Rebuttal follow-up**
> >
> > Dear Reviewer,
> >
> > We have worked hard to incorporate the review feedback by running new experiments and revising the paper. We'd really appreciate if you could confirm whether these changes address the concerns about the paper. If we have misunderstood any of the concerns, we'd like to learn that now so that we can further revise the paper or run additional experiments.
> >
> > Thanks!
> >
> > The Authors

---

> > > ### Author Response · Authors · 2024-11-30
> > > **Rebuttal follow-up**
> > >
> > > Dear Reviewer,
> > >
> > > We have worked hard to incorporate the review feedback by running new experiments and revising the paper. **Do the revisions and discussions above address your concerns?** We would greatly appreciate your engagement.
> > >
> > > Thanks!
> > >
> > > The Authors

---

> > ### Comment · Reviewer_h6Gn · 2024-12-03
> > **Response to Authors**
> >
> > Thank you for your response.
> >
> > Thanks for the update for the undefined symbols. This means that my original assessment is correct, and I didn't miss anything. I also noticed that the definitions of several key symbols, which are used in major results such as Theorem 1, 2, and Definition 3, are missing in the original submission. I'm curious as to why they were omitted.
> >
> > Additionally, the authors did not address my concerns regarding the confusing use of the terms "metric" and "quasimetric." This could make the paper difficult for readers to understand.
> >
> > Thank you for addressing how this work could inspire future research. It would be beneficial if the authors could present stronger results in the next version, as it would significantly enhance the paper's impact.

---

> ### Author Response · Authors · 2024-12-04
> **Additional Response**
>
> Thank you for your response.
>
> > Thanks for the update for the undefined symbols. This means that my original assessment is correct, and I didn't miss anything. I also noticed that the definitions of several key symbols, which are used in major results such as Theorem 1, 2, and Definition 3, are missing in the original submission. I'm curious as to why they were omitted.
>
> Due to space constraints, we initially moved some definitions to the appendix and omitted some standard definitions from prior work (for instance, the discounted state occupancy $p\_\gamma^\pi$ notation you noticed is used in numerous prior works [1,2,3,4,5]; similarly the goal-conditioned reward notation is defined in [6,7,8]).
>
> We now include these definitions in the main text to make the paper more self-contained. We hope the current revision addresses this concern.
>
> > Additionally, the authors did not address my concerns regarding the confusing use of the terms ''metric'' and ''quasimetric.'' This could make the paper difficult for readers to understand.
>
> The formal construction of the distances in this paper all rely on the quasimetric definition, which relaxes the more familiar notion of a metric by allowing asymmetric distances. While this notion of asymmetry is important in many MDPs, the key benefit of viewing things as (quasi)metrics is the triangle inequality, which is what enables planning invariance and horizon generalization. We use the term ''metric'' in some places when we wish to highlight the triangle inequality property rather than asymmetry, in line with prior work that formally relies on quasimetrics [9,10]. Note that since the definition of a metric is stronger than that of a quasimetric, all our results hold for metrics as well. We hope this note as well as our latest revision clarify this point.
>
> > Thank you for addressing how this work could inspire future research. It would be beneficial if the authors could present stronger results in the next version, as it would significantly enhance the paper's impact.
>
> Thank you for this suggestion. We can include results in additional AntMaze layouts [11] and the Humanoid environment [12] in our next revision. Future work may also benefit from evaluation on the newly-released OGBench baseline [13], which features a diverse set of long-horizon tasks in the *offline* setting.
>
> ---
>
> ### References
>
> [1] Eysenbach, B. et al., 2021. ''C-Learning: Learning to Achieve Goals via Recursive Classification.'' *ICLR*
>
> [2] Choi, J. et al., 2021. ''Variational Empowerment as Representation Learning for Goal-Conditioned Reinforcement Learning.'' *ICML*
>
> [3] Myers, V. et al., 2024. ''Learning to Assist Humans Without Inferring Rewards.'' *NeurIPS*
>
> [4] Schroecker, Y. and Isbell, C., 2020. ''Universal Value Density Estimation for Imitation Learning and Goal-Conditioned Reinforcement Learning.'' arXiv:2002.06473
>
> [5] Hoang, C. et al., 2021. ''Successor Feature Landmarks for Long-Horizon Goal-Conditioned Reinforcement Learning.'' *NeurIPS*
>
> [6] Fang, K. et al., 2022. ''Generalization With Lossy Affordances: Leveraging Broad Offline Data for Learning Visuomotor Tasks.'' *CoRL*
>
> [7] Yang, R. et al., 2023. ''What Is Essential for Unseen Goal Generalization of Offline Goal-Conditioned RL.'' *ICML*
>
> [8] Ghosh, D. et al., 2019. ''Learning Actionable Representations With Goal Conditioned Policies.'' *ICLR*
>
> [9] Liu, B. et al., 2023. ''Metric Residual Network for Sample Efficient Goal-Conditioned Reinforcement Learning.'' *AAAI*
>
> [10] Myers, V. et al., 2024. ''Learning Temporal Distances: Contrastive Successor Features Can Provide a Metric Structure for Decision-Making.'' *ICML*
>
> [11] Fu, J. et al., 2021. ''D4RL: Datasets for Deep Data-Driven Reinforcement Learning.'' arXiv:2004.07219
>
> [12] Gu, X. and Wang, Y., 2024. ''Advancing Humanoid Locomotion: Mastering Challenging Terrains With Denoising World Model Learning.'' *RSS*
>
> [13] Park, S. et al., 2024. "OGBench: Benchmarking Offline Goal-Conditioned RL." arXiv:2410.20092

---

### Official Review · Reviewer_7Gwa · 2024-11-04

**Soundness:** 1
**Presentation:** 2
**Contribution:** 2
**Rating:** 3
**Confidence:** 4

**Summary:**

This paper studies the generalization of RL from the perspective of horizon invariance. An RL agent that has been trained to reach nearby goals should succeed on distant goals and hence generalize to different horizon lengths. The paper intends to study how the techniques on invariance and generalization from other machine learning domains can be adapted to make goal-directed policies generalize to different horizons.

**Strengths:**

1. The paper aims to study an important question of generalization and planning under different horizon lengths for a goal-conditioned RL agent.
2. Connections between existing methods and the new proposed methods are provided.

**Weaknesses:**

1. The writing of the paper needs some work to strengthen the motivation and claims. The related works section can also benefit from further elaboration and coverage of the related literature.
2. The paper mentions that horizon generalization means that a policy that can achieve a goal n steps away should also be able to reach any new goal for which that original goal is a waypoint. This is a bit far-fetched of a statement as in a longer horizon, a lot of different things could happen over this longer timeframe and not all en-route waypoints would be relevant. Moreover, things could demand a completely different set of actions or approach after these n steps. Defining horizon generalization in this scenario needs a more thorough definition.
3. The example of dominoes considered in the paper is a very simple bandit setting, while the paper focuses on more general RL settings with different horizon lengths. Providing relevant working examples will be beneficial to the understanding.
4. Experiments are over simpler environments / task settings.

**Questions:**

1. Generalization in RL and in goal-conditioned setting can come in different forms. While generalizing over different horizons is important, why do you think invariance will help here? As the horizon changes and the trajectories get longer, it is possible that any of the waypoints relevant over shorter horizons are not that relevant anymore if better paths are discovered for example.
2. How does the method relate to when we are no longer looking at smaller horizons as being sub-goals of the longer horizons? Or even when we are not considering the goal as a condition. Do you think this work can include those cases?

---

> ### Author Response · Authors · 2024-11-22
> **Rebuttal**
>
> Dear Reviewer,
>
> Thank you for your detailed review and suggestions.
>
> Your primary concerns seem to be (1) issues with the motivation and formalism for planning invariance and horizon generalization and (2) concerns about the utility of our formalism in practical environments.
>
> We have made revisions based on your suggestions **highlighted in red** to clarify the theoretical results, added discussion on concrete takeaways to our conclusions, and updated our experiments to support our theoretical claims.
>
> **Do these changes fully address your concerns?** We believe the current claims in the paper are correct (regarding “this is a bit far-fetched”) and have clarified the proofs in Appendix B. If you still believe any of the proofs or other writing is unclear, please inform us and we will be happy to make further revisions.
>
> > Generalization in RL and in goal-conditioned setting can come in different forms. While generalizing over different horizons is important, why do you think invariance will help here?
>
> In ML, there is a strong connection between generalization and invariance where enforcing invariance facilitates generalization. For example, neural networks invariant to rotation show better abilities to generalize over rotated test sets [1, 2, 3]. This body of work motivates our analysis; we have revised Section 4.1 to refer to such work.
> We expect generalization over different horizons in RL to also relate to a form of invariance, and identify planning invariance as a possible mechanism to achieve horizon generalization.
>
> > How does the method relate to when we are no longer looking at smaller horizons as being sub-goals of the longer horizons?
>
> In general, *any* goal reached at a given horizon will necessarily have possible shorter-horizon subgoal(s) just by taking a distribution of states reached ``on the way’’ to the goal. This is mathematically formalized through our induction argument, which necessarily assumes that you can use smaller problems to solve larger problems.
> Does this clarify your concern? We are not entirely sure if we have understood it correctly, but would be happy to provide further information if needed.
>
> > Or even when we are not considering the goal as a condition. Do you think this work can include those cases?
>
> Outside the goal-conditioned setting, the distance metric in its current form loses meaning and we lose the benefits of ``reward-free” goal-conditioned reinforcement learning; we have added discussion of prior work and motivation for goal-conditioned RL in Section 2.
> However, our result suggests that if you force your reward structure to take on values that lead to quasimetric Q-functions, or Q-functions that are scaled and shifted versions of quasimetrics, then we should expect horizon generalization in goal-free RL. How to enforce such constraints on reward functions in non goal-conditioned settings is still an open question, but could have potential implications on how to engineer rewards (that correspond to a desired objective, even if it isn’t a fixed goal) to achieve horizon generalization.
>
> > The writing of the paper needs some work to strengthen the motivation and claims. The related works section can also benefit from further elaboration and coverage of the related literature.
>
> As suggested, we have made several revisions to make the paper more clear and add motivation. We have (1) expanded our prior works to compare horizon generalization to other forms of generalization with latent space abstractions (see Section 2) and motivated the goal-conditioned setting, (2) clarified the limitations of planning invariance as a mechanism to achieve horizon generalization (Section 4.2 and 4.5), (3) added a remark to highlight that horizon generalization, as defined in this paper, is nontrivial (Remark 2), and (4) replaced the dominoes setting example with a more complex navigation task to build intuition for planning invariance (Figure 2).
>
> > Experiments are over simpler environments / task settings.
>
> We would like to note that the primary Ant environment studied involves 8DoF control and 29d observations [4], and is challenging enough to have been used as a primary evaluation for several recent RL works [5,6,7,8]. Since the submission, we have expanded the results in the Ant environment to study generalization to quantitatively more distant goals, and have additionally run preliminary experiments in the Humanoid [4] environment showing similar trends, which will be included in the final version of the paper along with the existing results.

---

> > ### Author Response · Authors · 2024-11-22
> > **Rebuttal (cont.)**
> >
> > > As the horizon changes and the trajectories get longer, it is possible that any of the waypoints relevant over shorter horizons are not that relevant anymore if better paths are discovered for example.
> >
> > If the planning-invariant policy can navigate between any two pairs of nearby states with *complete state coverage* perfectly, then we expect these waypoints to remain relevant for longer horizons. However, there are settings where this complete state coverage assumption no longer holds and selected waypoints may not necessarily be optimal. We address these limitations through revisions in Section 4.5 and in the rest of this response. Our proofs show that waypoints will remain relevant for any long-horizon path, as waypoints consist of any intermediate state on the way to the goal. If an optimal waypoint is unseen, the policy will return suboptimal trajectories for longer horizon goals composed of shorter trajectories from the *covered* states (which may not be the true optimal waypoints). This is expected in the offline setting—if the optimal waypoint is never seen, whether as part of a short or a long trajectory in the training set, then it is challenging to learn a policy that navigates towards the true optimal waypoint.
> >
> > ## References
> >
> > [1] Cohen, T., Welling, M., 2016. "Group Equivariant Convolutional Networks." *ICML*
> >
> > [2] Benton, G. et al., 2020. “Learning Invariances in Neural Networks.” *NeurIPS*
> >
> > [3] Rowley, H., Baluja, S., Kanade, T., 1998. “Rotation Invariant Neural Network-Based Facial Recognition.” *IEEE*
> >
> > [4] Tassa, Y. et al., 2018. "DeepMind Control Suite." arXiv:1801.00690
> >
> > [5] Zheng, C. et al., 2023. "Contrastive Difference Predictive Coding." *ICLR*
> >
> > [6] Fujimoto, S. et al., 2018. "Addressing Function Approximation Error in Actor-Critic Methods." *ICML*
> >
> > [7] Park, S. et al., 2023. "HIQL: Offline Goal-Conditioned RL With Latent States as Actions." *NeurIPS*
> >
> > [8] Ghugare, R. et al., 2024. "Closing the Gap Between TD Learning and Supervised Learning--a Generalisation Point of View." *ICLR*

---

> > > ### Author Response · Authors · 2024-11-25
> > > **Rebuttal follow-up**
> > >
> > > Dear Reviewer,
> > >
> > > We have worked hard to incorporate the review feedback by running new experiments and revising the paper. We'd really appreciate if you could confirm whether these changes address the concerns about the paper. If we have misunderstood any of the concerns, we'd like to learn that now so that we can further revise the paper or run additional experiments.
> > >
> > > Thanks!
> > >
> > > The Authors

---

> > > > ### Author Response · Authors · 2024-11-30
> > > > **Rebuttal follow-up**
> > > >
> > > > Dear Reviewer,
> > > >
> > > > We have worked hard to incorporate the review feedback by running new experiments and revising the paper. **Do the revisions and discussions above address your concerns?** We would greatly appreciate your engagement.
> > > >
> > > > Thanks!
> > > >
> > > > The Authors

---

### Author Response · Authors · 2024-11-26
**Rebuttal follow-up**

Dear AC,

**Could you kindly ask the reviewers to read the rebuttal and let us know if there are remaining suggestions for the paper?** We have worked hard to address the reviewer feedback through new experiments and revisions. As tomorrow is the last day for revisions, we'd like to make sure that there aren't any additional suggestions.

Kind regards,

The Authors

---

### Meta-Review · Area_Chair_qUqJ · 2024-12-18

**Metareview:**

This paper studies the properties of policies that generalize with respect to horizon, which the authors highlight is linked to the notion of _invariance to planning_. The authors provide theoretical analyses on the realizability of horizon generalization and planning invariance, and combine it with an empirical evaluation that serves as evidence for their theoretical findings.

The main weakness the reviewers found with the work was in its utility and downstream applicability, as well as in its clarity. During the rebuttal period the authors provided ample revisions to address the reviewer concerns, but unfortunately most reviewers were rather unresponsive.

I went through the paper myself and found it well-written and well-motivated, in particular considering the modifications made during the rebuttal. The main weakness I find is in the exposition of the experiments in section 6, as the wording is sometimes a little confusing to follow. I recommend the authors revise the wording of this section carefully. For example, in the caption in Figure 4 it says "combining that policy with planning does not increase performance", which is not true; what's true is that it increases _less_, but there is an improvement.

Overall, however, I believe these corrections are rather minor at this point, and I found the paper to be a good submission under my own revision.

Thus, given my reading of the paper, the unresponsiveness of the reviewers to a thorough rebuttal to the authors, I am recommending an acceptance (against the overall sentiment of the reviewers).

**Additional Comments On Reviewer Discussion:**

All reviewers provided a good initial review, to which the authors provided thorough rebuttals (which, to my reading, addressed all concerns). Most of the reviewers were not responsive (or were not confident in their responses), which prompted me to review the paper myself.

Upon doing so, I am inclined to recommend acceptance.

---

### Decision · Program_Chairs · 2025-01-22

Accept (Poster)